# Signal response of the Swiss plate geophone monitoring system impacted by bedload particles with different transport modes

Zheng Chen[1,2,3], Siming He[1], Tobias Nicollier[2], Lorenz Ammann[2], Alexandre Badoux[2], Dieter Rickenmann[2]

[1]Institute of Mountain Hazards and Environment, Chinese Academy of Sciences, Chengdu, 610041, China
[2]Swiss Federal Institute for Forest, Snow and Landscape Research WSL, Birmensdorf, 8903, Switzerland
[3]University of Chinese Academy of Sciences, Beijing, 100049, China

*Correspondence to*: Zheng Chen (zheng.chen@wsl.ch; chenzcas@gmail.com)

**Abstract.** Controlled experiments were performed to investigate the acoustic signal response of the Swiss plate geophone (SPG) system impacted by bedload particles varying in size, impact angle and transport mode. The impacts of bedload particles moving by saltation, rolling, and sliding were determined by analyzing the experimental videos and corresponding vibration signals. The finite element method (FEM) was utilized to construct a numerical model of the SPG system and to simulate the signals triggered by a quartz sphere hitting the plate with impact angles ranging from 0° to 90°. For a particle impact on the bed or on the geophone plates, the signature of the generated signal in terms of maximum amplitude, number of impulses and centroid frequency was extracted from the raw monitoring data. So-called signal packets were determined by performing a Hilbert transform of the raw signal. The number of packets was calculated for each transport mode and for each particle size class, with sizes ranging from 28.1 mm to 171.5 mm. The results show how the number of signal impulses per particle mass, the amplitude of the signal envelope, and the centroid frequency change with increasing particle size, and they also demonstrate the effect of bedload transport mode on the signal response of the SPG system. We found that there is a general increase in the strength of the signal response or in the centroid frequency when the transport mode changes from sliding to rolling to saltation. The findings of this study help to better understand the signal response of the SPG system for different bedload transport modes, and may also contribute to an improvement of the procedure to determine bedload particle size from the SPG signal.

## 1 Introduction

Quantification of bedload transport processes constitutes a significant challenge in river dynamics and can provide a prerequisite for the design of hydraulic engineering structures and for the assessment of natural hazards (Rickenmann, 2016). Additionally, measurements of bedload transport rates in both laboratory and field help to improve understanding of its transport mechanism and to validate existing models or formulas (Habersack and Laronne, 2002; Schneider et al., 2015; Rickenmann, 2020).

In general, there are two types of methods for measuring bedload transport rate, including (1) direct methods to measure the transported bedload mass, installing physical samplers and traps on the river bed for some time frame (Bunte et al., 2004;

Childers, 1999; Emmett, 1980; Hayward, 1980; Helly and Smith, 1971; Gray et al., 2010; Ryan et al., 2005); (2) indirect methods, in particular, the acoustic-based monitoring devices, including piezoelectric sensors (Krein et al., 2008; Rickenmann and McArdell, 2007), hydrophones (Barton et al., 2010; Camenen et al., 2012; Rigby et al., 2015, 2016; Geay et al., 2017; Geay et al., 2020), ADCP (Acoustic Doppler Current Profiler) (Rennie et al., 2017; Conevski et al., 2019), pipe microphones (Mizuyama et al., 2010), geophones (Rickenmann et al., 2012, 2014; Rickenmann, 2017) and seismic sensors (Bakker et al., 2020; Farin et al., 2019; Gimbert, 2019; Roth et al., 2017; Tsai et al., 2012).

The advantage of the indirect bedload measuring method is to provide long-term continuous data on bedload transport (Rickenmann, 2017). In comparison, the direct bedload measuring method is suited for gravel-bed streams under the condition of low- or medium-discharge levels and typically relatively short sampling duration (Gray et al., 2010), and taking bedload samples can be challenging in case of large flow discharges and steep streams (Rickenmann and Fritschi, 2017; Nicollier et al., 2019). However, the indirect measurements must be calibrated using the direct methods (Wyss et al. 2016a, 2016b).

Acoustic-based indirect devices record the vibration signals generated by bedload particles impacting on a stream bed, an impact plate, or an impact pipe (Rickenmann, 2017). The acoustic vibration signal contains information, e.g. in terms of amplitude, impulses and characteristic frequency (Barton et al., 2006; Burtin et al., 2008, 2010, 2011; Govi et al., 1993; Hsu et al., 2011; Tsai et al., 2012; Vasile, 2020), which can be used to infer the bedload transport rates (Wyss et al., 2016a, 2016b; Nicollier et al., 2020). One such method, the Swiss plate geophone (SPG) system is a robust monitoring device that records the acoustic signal generated by bedload particle impacting onto steel plates. The SPG system was deployed at more than 20 field sites, mainly across Europe (Rickenmann, 2017), aiming to derive bedload fluxes and particle size distributions (Wyss et al., 2016c). Significant differences between field-based calibration relationships were found to be possibly caused by variations of particle impact location and impact angle (Turowski et al., 2013), particle shape (Cassel et al., 2021; Krein et al., 2008), streamflow velocity (Rickenmann et al., 2014; Wyss et al., 2016a), grain size distribution (Nicollier et al., 2021a), and particle transport mode (Krein et al., 2008; Turowski and Rickenmann, 2009; Turowski et al., 2015).

In addition to field calibration measurements, controlled flume experiments were conducted with different types of acoustic devices (Beylich and Laute, 2014; Moen et al., 2010; Mizuyama et al., 2010; Wyss et al., 2016a), to investigate their suitability for monitoring of bedload transport with variable transport modes (Tsakiris et al., 2014). In particular, the bedload transport modes, namely, saltation, rolling and sliding, influence the acoustic signal response of geophones or other acoustic sensors (Tsakiris et al., 2014), thus affecting the signal-bedload calibration relations. Previous studies have shown that the transported particles in the mode of rolling and sliding are associated with a more important signal power at lower frequencies compared to salting particles (Krein et al., 2008). This finding was also supported by a flume experiment with a geophone impact plate, using unisize spherical glass beads with different transport modes (Tsakiris et al., 2014). Additionally, the signal responses of the geophone were observed to depend both on flow conditions and on transport modes (Tsakiris et al., 2014). It is therefore important to quantify the effect of the transport mode on the signal response, as this will eventually affect the signal-based particle size classification.

Controlled outdoor flume experiments with the SPG system (Nicollier et al., 2021a) were carried out to better understand the influence of transport mode. However, flow conditions (turbidity, illumination) sometimes impaired the clarity of the videos that were recorded by a camera during the experiments to capture the motion characteristics of bedload particles. A way to extend the experimental data is to apply the finite element method (FEM), which has been used already successfully to simulate the structural dynamic responses of the SPG system impacted by a quartz sphere falling vertically onto the SPG plate (Chen et al., 2021). The dynamic response of the SPG system that corresponds to the recorded signal can be fully described by the partial differential equations (PDEs) based on elastoplastic mechanics, and these PDEs can be numerically solved by the FEM formulations resulting in a system of algebraic equations. The FEM simulation is used here also for non-vertical impacts to investigate the effect of different bedload impact angles, covering a wide range of angles for transport modes (saltation, rolling, and sliding) observed in the flume experiments.

The aim of the present paper is to investigate how the signal response of the SPG system impacted by bedload particles changes for different transport modes. First, controlled flume experiments and inclined chute experiments were performed with natural bedload particles and quartz spheres. On the basis of the video material recorded during these experiments, we compared for each impact event the motion of the bedload particles, including transport mode, impact position, and impact instant with the acoustic signal recorded by the SPG system. Second, a FEM model of the SPG system was used to simulate the signal response of the SPG system produced by quartz spheres with varying impact angles ranging from 0° to 90° for different particle sizes, and the results were compared with the observations from the inclined chute experiment. Data from the physical experiments and the numerical simulations were analyzed quantitatively in terms of signal responses for each transport mode and for changing particle size.

## 2 Methods

In the methods section, we introduce in turn the controlled experiments including controlled flume experiments and inclined chute experiments, numerical simulations with the FEM model, methods of transport mode analysis and signal processing.

### 2.1 Controlled experiments

#### 2.1.1 Experimental set-up

Full-scale controlled flume experiments were performed with natural bedload particles varying in size (Nicollier et al., 2019, 2020, 2021a), using an outdoor experimental facility at the Oskar von Miller Institute of the Technical University of Munich in Obernach, Germany. The entire experimental system can be divided into serval parts including the flume channel made of concrete, the measuring reach equipped with different types of sensors (**Fig. 1a**), namely the Swiss plate geophone (SPG) system, the miniplates accelerometer (MPA) and the Japanese pipe microphone (JPM), and the basin for collecting and recycling bedload particles. This experimental system enables quantitative investigations regarding the process of bedload transport, observing the characteristics of the particles motion and measuring the vibration signals during the bedload transport

process. The experimental channel reach used in this study has a rectangular cross-section, a length of 24 m, a width of 1.02 m, a maximum depth of 2.02 m and a slope of 4% (Nicollier et al., 2019). The channel bed roughness is made up by gravel particles that have a size corresponding to $D_{67}$ and $D_{84}$ (see **Tab. 1**) of the bedload material sampled at the Navisence field site in Switzerland, embedded in concrete and about half their size protruding into the flow. The SPG system is installed in the measuring reach (**Fig. 1a**) with the plates mounted flush with the channel bed and with the geophone sensor recording the

vertical vibration (displacement velocity) of the plate. The side wall of the measuring reach is made up by plexiglass for video observation.

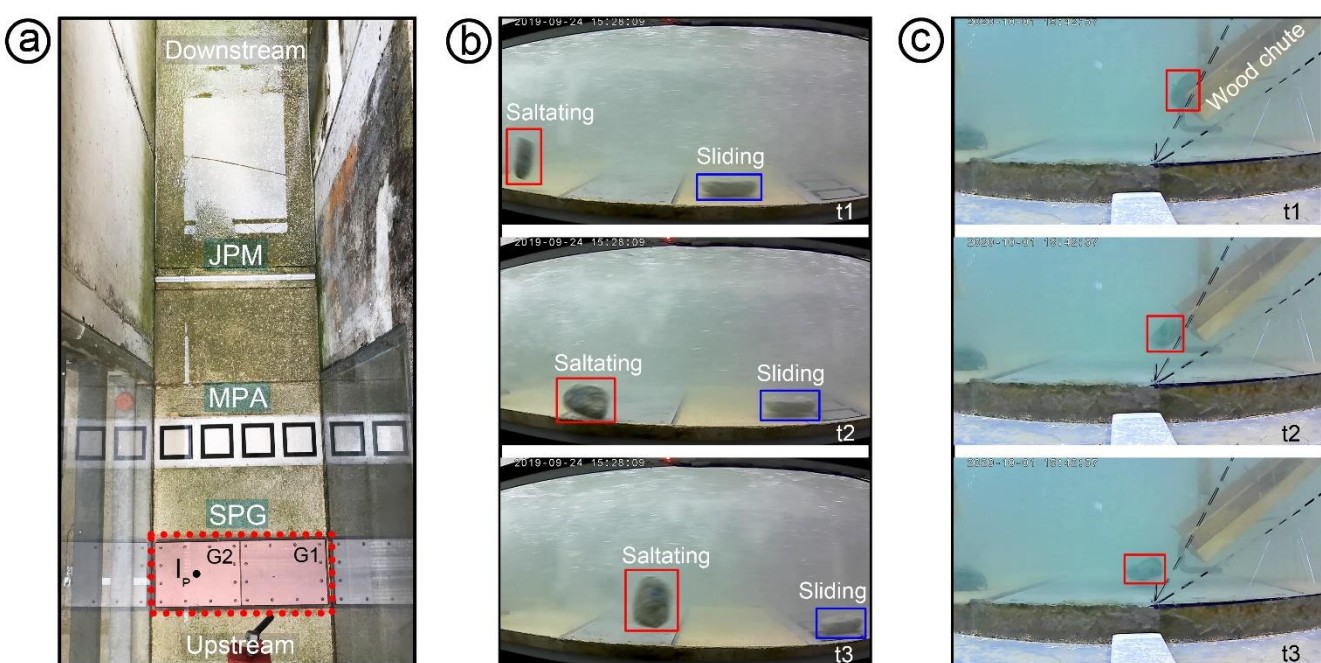

**Figure 1:** Measuring site at the Obernach experimental flume. (a) Measuring reach with different types of sensors mounted on the flume bed, including the Swiss plate geophone (SPG), the miniplate accelerometer (MPA) and the Japanese pipe
microphone (JPM). (b) Frames from video recorded during a controlled flume experiment, used for the tracking of particles (mean b-axis = 127.9 mm) impacting onto and moving over the SPG plates. The particle marked with the red rectangle is transported in saltation, while the one marked with the blue rectangle is sliding. (c) Frames from video recorded during a drop experiment with a wood chute inclined at an angle of 45°. G1 and G2 in (a) are two plates of the SPG system, and the black dot $I_P$ marks the impact location of the bedload particles on the plate G2 for the inclined chute experiments. The time interval
between consecutive frames for each column in (b) and (c) is 1/3 s. The length of the geophone plate in flow direction is 0.36 m.

### 2.1.2 Flume experiments

During the flume experiments, the flow velocity and flow depth were adjusted to match with that at the Navisence field site. The experimental flow rate was maintained constant and the flow was roughly uniform along the experimental reach, with a
115 flow depth of about 0.54 m and the flow velocity set to 3.3 m/s and monitored using a flow meter (OTT MFpro) positioned 0.1 m above the SPG plate in the middle of the cross-section. Information on bed characteristics and hydraulic conditions are

given in **Tab. 1**. The bedload particles with a natural shape were released into the flume several meters upstream of the SPG system. A Lenco camera was set in a side view perpendicular to the plexiglass side wall to record videos with 30 frames per second (FPS) throughout the duration of each experiment. **Fig. 1b** shows typical images of two different particles of size class

C9, moving over the SPG plates. The video recordings were analyzed frame by frame and the instants of bedload particle impacts on the concrete bed and the SPG plates were determined. In addition, the transport modes of the particle were assessed from the videos (i.e. saltation, rolling or sliding, as illustrated in **Fig. 3**). The experimental particles were sorted into 10 size (C1 to C10) classes ranging from 12.3 mm to 171.5 mm (**Tab. 2**). The averaged bedload fluxes for all particle size classes were estimated ranging from 0.092 kg/m/s to 20.31 kg/m/s (see also **Tab. 2**). In this study, only the data obtained from

experiments involving the particle size class C4 to C10 are presented. Particle impacts for the size classes C1 to C3, ranging from 12.3 mm to 21.8 mm, were difficult to distinguish in the videos due to (1) poor lighting conditions resulting in low contrast in the video frame image, and (2) the large number of small-sized particles used in each experimental run.

**Table 1:** Bed and flow conditions at the Navisence field site and in the flume experiments.

| Parameters | Units | Value |
|---|---|---|
| Bed surface $D_{67}$ | mm | 180 |
| Bed surface $D_{84}$ | mm | 280 |
| Flow depth (Navisence) | m | 0.4-0.65 |
| Flow depth over the SPG (flume) | m | 0.54 |
| Flow discharge (flume) | $m^3 \, s^{-1}$ | 1.78 |
| Flow discharge (Navisence) | $m^3 \, s^{-1}$ | 1.2-2.28 |
| Flow velocity (Navisence) | $m \, s^{-1}$ | 3-3.5 |
| Flow velocity 0.1 m above the SPG plates (flume) | $m \, s^{-1}$ | 3.30 |
| Flume gradient of the natural bed | % | 4 |
| Flume width | m | 1.02 |
| Froude number (flume) | - | 1.43 |
| Froude number (Navisence) | - | 1.39-1.51 |

**Table 2:** Bedload particle characteristics for each grain size class $j$.

| Bedload size class $j$ [-] | Mean size $D_j$ [mm] | Mean particle mass $M_j$ [kg] | Number of particles for each run $n$ [-] | Averaged bedload flux [kg s $m^{-1}$] |
|---|---|---|---|---|
| C1 | 12.3 | 0.003 | 50 | 0.092 |
| C2 | 17.4 | 0.010 | 50 | 0.218 |
| C3 | 21.8 | 0.019 | 40 | 0.411 |
| C4 | 28.1 | 0.041 | 33 | 0.708 |
| C5 | 37.6 | 0.094 | 20 | 0.792 |
| C6 | 53.2 | 0.265 | 20 | 1.824 |
| C7 | 71.3 | 0.574 | 20 | 5.764 |
| C8 | 95.5 | 1.249 | 10 | 6.907 |
| C9 | 127.9 | 3.633 | 5 | 10.43 |
| C10 | 171.5 | 8.743 | 5 | 20.31 |

### 2.1.3 Inclined chute experiments

Significant differences between transport modes (saltation, rolling, and sliding) were observed with regard to the impact angle on the channel bed. Therefore, an inclined chute experiment was conducted in still water to examine the effect of particle impact angle on the signal response of the SPG system (**Fig. 1c**). The length of the chute was 1.0 m and the width was about 0.1 m. Due to the solid friction it was difficult for the particles released at the top of the wood chute to keep moving at small chute angles. Hence, the experimental angles in this study were chosen as 45° and 60° for natural bedload particles with sizes ranging from 12.3 mm to 95.5 mm (where the size is given as the b-axis of the particle) and for spherical particles with sizes ranging from 20 mm to 82 mm (see **Tab. 3**). For each test, the flow velocity was around 0 m/s (no flow) and the water depth was 0.84 m. The impact velocity of the bedload particle on the SPG plates was determined to be about 3.7 m/s and 4.1 m/s for chute angles of 45° and 60°, respectively, considering the energy conservation law or estimated using the experimental video frames. Note that the impact velocity in the inclined chute experiments is considerably higher than the average impact velocities of the particles in the flume experiments that are generally estimated as fractions of a meter per second, and the investigated angles are rather steep compared to another study with smaller experimental particles (Auel et al. 2017b). The inclined chute experiments were performed in this study just to investigate the effect of impact angle on the SPG signal responses and to compare with the results obtained by numerical model introduced below.

**Table 3:** Mean particle size $D_j$ and mass $M_j$ and number of test repetitions $m$ for bedload particle size class $j$ for the impact experiments with channel angles of 45° and 60°. S1, S2, S3, and S4 refer to four quartz spheres of increasing size.

| Bedload size class $j$ [-] | Mean size $D_j$ [mm] | Mean mass $M_j$ [kg] | Number of tests $m$ [-] | Chute slope angle $\theta$ [°] |
|---|---|---|---|---|
| C1 | 12.3 | 0.003 | 10 | 45°, 60° |
| C2 | 17.4 | 0.010 | 10 | 45°, 60° |
| C3 | 21.8 | 0.019 | 10 | 45°, 60° |
| C4 | 28.1 | 0.041 | 10 | 45°, 60° |
| C5 | 37.6 | 0.094 | 10 | 45°, 60° |
| C6 | 53.2 | 0.265 | 10 | 45°, 60° |
| C7 | 71.3 | 0.574 | 10 | 45°, 60° |
| C8 | 95.5 | 1.249 | 10 | 45°, 60° |
| S1 | 20.0 | 0.012 | 5 | 45°, 60° |
| S2 | 31.0 | 0.050 | 5 | 45°, 60° |
| S3 | 51.0 | 0.185 | 5 | 45°, 60° |
| S4 | 82.0 | 0.760 | 5 | 45°, 60° |

### 2.2 Numerical simulations

To supplement the experimental data, particularly for smaller impact angles, a finite element method (FEM) was built to produce a virtual model of the SPG system, as illustrated in **Fig. 2.** The FEM model includes structural components of the SPG system, such as the steel plate, bolts, sensor casings, elastomers, and the internal and outer frames were subdivided, individually, into small finite elements. Subsequently, all the components were assembled considering mechanical contacts and frictions,

and the entire SPG system was simulated in the LS-DYNA (LSTC 2014). Detailed information used in the FEM model are reported by Chen et al. (2021). Before the numerical simulations, the FEM model has been calibrated with results obtained from the previous lab experiments (drop tests) with quartz spheres (see **Appendix A**, also see Chen et al. 2021). The FEM

model was used to numerically simulate the signal response for spherical particles impacting a SPG plate with a velocity of 3.5 m/s (irrespective of the impact angle) at different angles ranging from 0° to 90°, and for sphere diameters of 82 mm, 95.5 mm, and 120 mm, as indicated in **Tab. 4**.

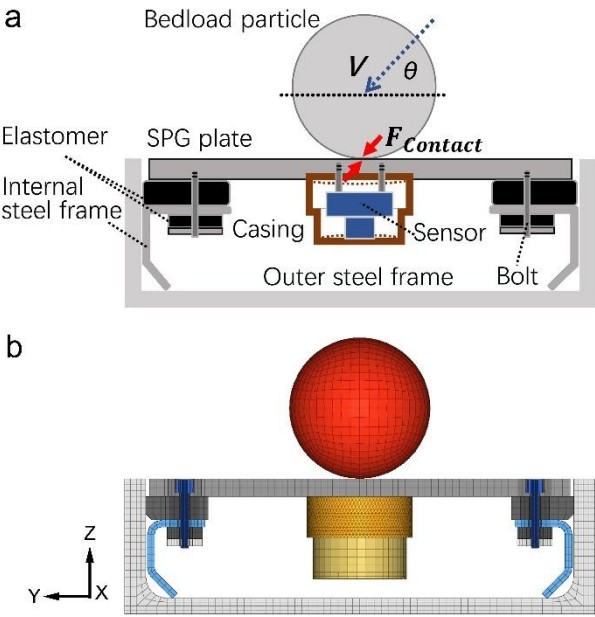

**Figure 2:** (a) Cross-sectional sketch of the SPG system impacted by a spherical particle. (b) Cross-sectional view of the FEM
model of the SPG system. The coordinate system is set up with the X-axis pointing in the transverse direction (across the flume width), the Y-axis pointing downstream (in flow direction), and the Z-axis pointing up perpendicularly to the plate's surface. $\theta$ is the impact angle of the sphere. $F_{Contact}$ is the contact force between the sphere and the plate. $V$ is the impact velocity of the sphere onto the plate, which has two components $V_Y$ and $V_Z$ given in Tab. 4.

**Table 4:** Characteristic values of the spheres and impact angles used in the FEM simulations. The diameters $D_1$, $D_2$ and $D_3$ of the spheres are 82 mm, 95.5 mm and 120 mm, respectively. A constant density $\rho_s = 2677$ kg/m$^3$ was used in the FEM simulations. $V_Y$ and $V_Z$ are the components of the impact velocity in the Y and Z directions, respectively.

| No. | $D_1$ [mm] | $D_2$ [mm] | $D_3$ [mm] | Impact angle [°] | $V_Y$ [m/s] | $V_Z$ [m/s] |
|---|---|---|---|---|---|---|
| 1 | 82.0 | 95.5 | 120.0 | 0 | 3.500 | 0.000 |
| 2 | 82.0 | 95.5 | 120.0 | 10 | 3.447 | 0.608 |
| 3 | 82.0 | 95.5 | 120.0 | 20 | 3.289 | 1.197 |
| 4 | 82.0 | 95.5 | 120.0 | 30 | 3.031 | 1.750 |
| 5 | 82.0 | 95.5 | 120.0 | 45 | 2.475 | 2.475 |
| 6 | 82.0 | 95.5 | \ | 60 | 1.750 | 3.031 |
| 7 | 82.0 | 95.5 | \ | 70 | 1.197 | 3.289 |

| 8 | 82.0 | \ | \ | 80 | 0.608 | 3.447 |
| 9 | 82.0 | \ | \ | 90 | 0.000 | 3.500 |

## 2.3 Bedload transport modes

### 2.3.1 Saltation, rolling and sliding

Generally, bedload particles are transported in three types of motions, namely saltation, rolling and sliding, as illustrated in **Fig. 3**. In other studies, the motion mode of the bedload transport was investigated experimentally and showed a correlation with the time-averaged bed shear stress $\overline{\tau_b}$ or the ratio of $\overline{\tau_b}$ to the critical value of the bed shear stress $\tau_{critical}$ for incipient particle motion (Tsakiris et al., 2014). The value of $\overline{\tau_b}$ is constant in the case of the uniform flow condition, which can be calculated as

$$\overline{\tau_b} = \rho g R_h S ,\qquad(1)$$

where $\rho$ is the water density, $g$ is the gravity acceleration, $S$ is the bed slope, and $R_h$ is the hydraulic radius that can be expressed as $bh/(2h + b)$ for a rectangular cross section, $h$ is the flow depth, and $b$ is the channel width. For our flume experiments, $\overline{\tau_b}$ is determined as 102.9 N/m$^2$.

The critical Shields parameter $\Theta_{Critical}$ is defined as the ratio of the critical bed shear stress $\tau_{critical}$ to the submerged particle
weight:

$$\Theta_{Critical} = \frac{\tau_{Critical}}{(\rho_s - \rho)gD} ,\qquad(2)$$

where $\rho_s$ is the particle density.

An estimation of $\Theta_{Critical}$ for our experimental conditions was made in two ways. First, $\Theta_{Critical}$ was estimated based on the maximum particle size $D_{Max} = 171.5$ mm transported in our experiments, assuming that this size is close to (but not equal to)
the critical size of bedload particles that started moving during the experiments. However, the true value of $\Theta_{Critical}$ should be somewhat smaller than the estimated value $\frac{\overline{\tau_b}}{(\rho_s - \rho)gD_{Max}} = 0.037$. Second, considering that the controlled experiments in this study were performed in a flume facility reconstructed from the field site, the critical Shields parameter $\Theta_{Critical}$ should be rather similar to that at the field site. According to the study of Schneider et al. (2015) including several mountain streams, the median value of the effective shear stress (corresponds to $\Theta_{Critical}$) has been determined to be about 0.03 from the main dataset, showing
less dependency with the slope of the stream bed. Shahmohammadi et al. (2021) statistically obtained $\Theta_{Critical}$ vs relative roughness correlation curves from the data of a large number of flume experiments. The relative roughness of our experiments ranges from 0.023-0.32, resulting in a median value for the critical Shields parameter of approximately 0.05. However, given our flume-based estimate and the fact that our experimental conditions are comparable to the filed sites investigated by Schneider et al. (2015), the critical Shields parameter $\Theta_{Critical}$ in our flume experiments is assumed as 0.03.
Then the excess transport stage $T$ (Auel et al., 2017a) can be calculated by **Eq. 3**:

$$T = \frac{\overline{\tau_b}}{\tau_{Critical}} - 1 = \frac{R_h S}{\Theta_{Critical}\left(\frac{\rho_s}{\rho}-1\right)D} - 1 \,, \tag{3}$$

Studies have shown that the probabilities of saltation $P_{Sal}$, rolling $P_{Rol}$, and sliding $P_{Sli}$ are related to the flow intensity or $T$ (Auel et al., 2017a; Hu and Hui, 1996a). For our flume experiments, $T$ is calculated ranging from 0.22 to 6.42 for the particle size ranging from 171.5 mm to 28.1 mm.

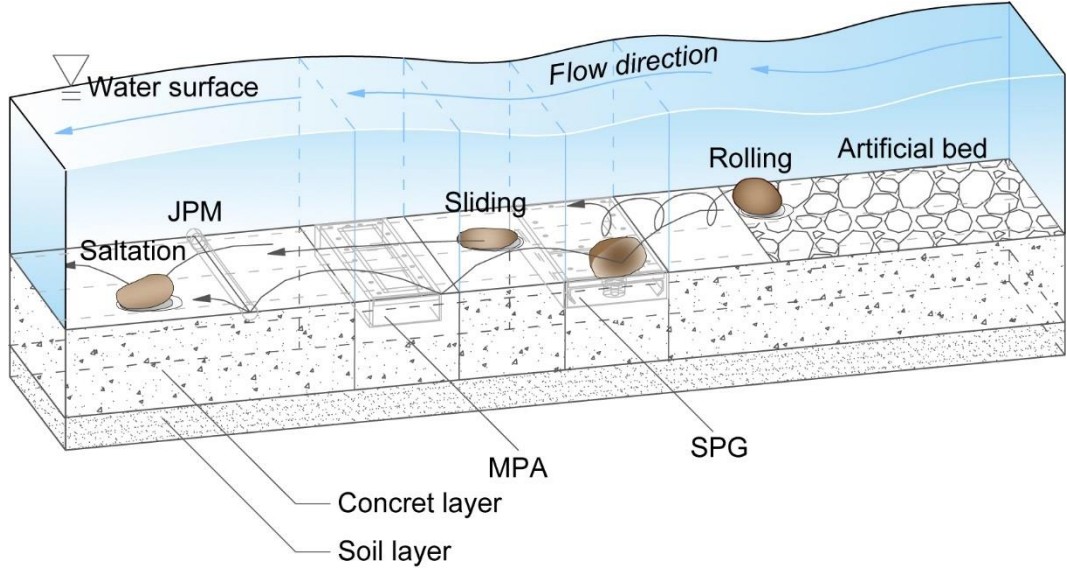

**Figure 3:** Sketch of bedload particles in different transport modes, including saltation, rolling and sliding, moving over the Obernach flume facility.

### 2.3.2 Impact instant and video analysis

In order to match the transport mode of a bedload particle with the vibration signal, an important parameter that needs to be
determined from experimental videos is the time instant when a particle impacts onto the channel bed. **Figs. 4a-4c** show sketches of transport modes of saltation, rolling and sliding, respectively, and also indicate an interaction between the bedload particle and the SPG plate. The forces introduced in following sketches (**Fig. 4**) are used only as an aid to illustrate how we observe a few moments when the particles are in contact with the plate or the channel bed. Specifically, a shear stress between the geophone plate and the contact surface of a particle is generated when the particle impacts onto the plate with a certain
angle, as seen in **Fig. 4a**. The frictional force $F_c$ together with the fluid drag force $F_w$ form a force couple. Similarly, another set of force couple is present in the vertical direction, namely the vertical support force $F_n$ and the particle weight force $G$. These force couples act together on the particle, and finally rotate the particle. This small rotation of the bedload particle occurs immediately after impacting, allowing to determine the impact instant (at $T_1$) from the video frames. **Appendix B** gives more details on how we analyzed the experimental videos.

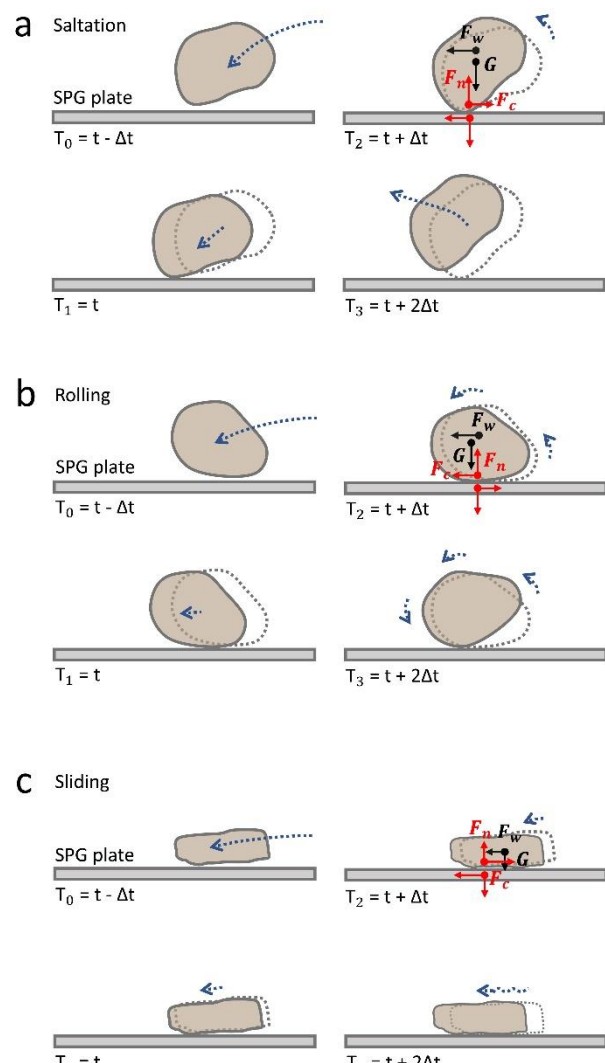

**Figure 4:** Schematic illustration of the three observed types of transport modes: (a) saltation, (b) rolling and (c) sliding. $T_0$, $T_1$, $T_2$, and $T_3$ are four different time instants of particle motion, indicating impact and rebound of a particle on the SPG plate. In particular, $T_2$ is the instant when the bedload particle impacts on the SPG plate. $F_c$ and $F_n$ denote a friction force and a vertical support force exerted by the SPG plate on the bedload particle, respectively. $F_w$ is a force of water acting on the bedload particle. $G$ is particle weight force.

**2.4 Signal processing**

**2.4.1 Signal characteristics: amplitude, impulse, frequency**

A typical signal response of the SPG system recorded during a flume experiment for the bedload particles of grain-size class C9 moving over the SPG plates is illustrated in **Fig. 5**. The packets (**Figs. 5a** and **5b**) were delimited on the basis of the envelope (blue line) of the signal computed with Hilbert transform (Wyss et al., 2016a). Each packet corresponds to the signal

response following a single particle impact onto the SPG plate, as seen in **Fig. 5c**. Subsequently, these packets were classified according to the respective transport modes of saltation (in gray), rolling (in red) and sliding (in blue), as determined from the experimental videos that were introduced above. The packets colored in purple suggest that the signals of this packet recorded by the sensor G1 or G2 were triggered by impacts on the neighboring sensor G2 or G1, respectively, or they represent the signals that cannot be matched with the videos due to limitation of light conditions.

The positive maximum amplitude of a packet is given as $Amp_{Max,\ Pac}$ (V), as seen in **Fig. 5c**. The number of impulses $I$ (**Fig. 5d**) of each packet is obtained by counting the times of positive signal excursions above the pre-defined system threshold (Rickenmann et al., 2012, 2014; Wyss et al., 2016a). The threshold value in our study is 0.0216 V, as indicated by the blue dash-dotted line in **Fig. 5d**. Based on field bedload measurements at various sites, the number of impulses $I$ has been found to be reasonably well correlated with the total transported bedload mass $M_{Tot}$, using the equation $I = k_b M_{Tot}$, where $k_b$ is the site-dependent calibration coefficient. The coefficient $k_b$ is further developed for different grain size classes $j$ as the coefficient $k_{bj}$, which has been utilized to infer bedload transport from the SPG signals by grain-size fractions. (Wyss et al., 2016c; Nicollier et al., 2020).

The mass-impulse coefficient $k_{IPM}$ used in the present study is similar to the coefficient $k_b$ in other studies (Rickenmann et al., 2014; Nicollier et al., 2021a) but more comparable to the $k_{bj}$ value, although not completely the same. $k_{IPM}$ was used as a parameter relating the signal impulses triggered by each impact to the transported bedload mass $M$ (Chen et al., 2021), and is defined as the number of impulses per particle mass:

$$k_{IPM} = \frac{I}{M}, \tag{4}$$

where $I$ is the number of signal impulses recorded by the SPG system and $M$ is the corresponding transported particle mass.

According to the Hertz theory, the centroid frequency $Freq_{Centroid}$ (**Eq. 5**) of the SPG signal is an important parameter that can help to support the bedload size identification (Wyss et al., 2016a; Rickenmann, 2017; Thorne, 2014):

$$Freq_{Centroid} = \frac{\sum f_m A_{FFT,m}}{\sum A_{FFT,m}}, \tag{5}$$

where $f_m$ is the spectrum frequency (Hz) and $A_{FFT,m}$ is the amplitude (V·s) that is obtained by performing fast Fourier transform FFT on the signals. Note that the definition of the centroid frequency in (Eq. 5) is different from the definition of the central frequency in Thorne (1986).

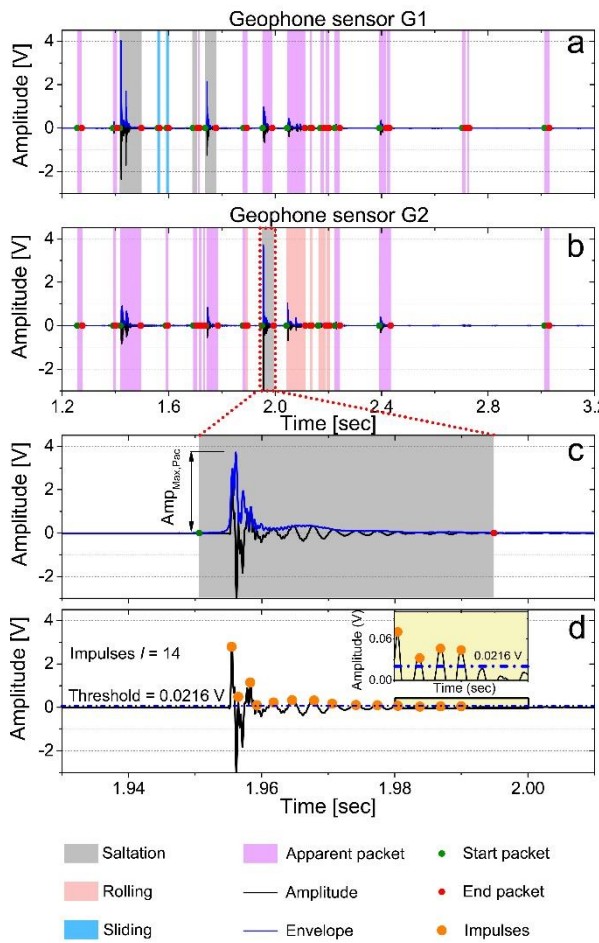

**Figure 5:** Illustration of the SPG vibrations and signal packets for different transport modes following a flume experiment with bedload particles of grain-size class C9. (a) and (b) represent signals that were recorded by geophone sensor G1 and G2, respectively, with a flow velocity of 3.3 m/s. (c) Illustration of the packet definition as the envelope (blue line) of the raw signal, computed with the Hilbert transform, and representing one impact of a saltating particle. (d) Definition of impulse counts $I$ (= 14), counting the times the signal exceeds the threshold (0.0216 V, see the blue dash-dotted line) in the positive domain.

### 2.4.2 Number of packets

The amplitude and frequency characteristics of the signal were found to vary significantly with the impact locations of the bedload particle, in particular when an impact occurs on a neighboring plate or on the concrete bed of the channel. An amplitude-frequency-based filtering method has been developed by Nicollier et al. (2021b, in review) to identify packets generated by these impacts and to classify them as "apparent". In contrast, packets generated by bedload particles impacting on the SPG plate above the considered geophone sensor are being classified as "real". This filtering process accounts for the phenomenon of attenuation acting on a propagating seismic wave. In fact, the further a seismic wave propagates, the stronger is the attenuation of high frequencies with regard to low frequencies and thus the lower is the energy of the wave. "Apparent"

packets can therefore be identified and removed from the final packet counting on the basis of their low amplitude-frequency content.

Subsequently, the ratio $r_{i,j}^{Packet,V\_F}$ of the total number of real packets over all transport modes based on the video observations to the real-packet number determined by the filtering method is calculated by

270 $$r_{i,j}^{Packet,V\_F} = \frac{N_{i,j}^{Packet,V}}{N_{i,j}^{Packet,F}}, \tag{6}$$

where $N_{i,j}^{Packet,V}$ is the total number of real packets for experimental run $i$ and grain-size class $j$ over transport modes based on the video analysis; $N_{i,j}^{Packet,F}$ is the number of real packets for experimental run $i$ and grain-size class $j$, determined by the filtering method.

In addition, similar to the definition in Wyss et al. (2016c), the ratio $\alpha_{i,j}^{Packet}$ of the number of packets $P_{i,j}$ to the number of 275 particles $N_{i,j}$ for each experimental run $i$ and grain-size class $j$ is given as:

$$\alpha_{i,j}^{Packet} = \frac{P_{i,j}}{N_{i,j}}, \tag{7}$$

For each transport mode, using all detected packets including both "real" and "apparent" packets, we have:

$$\alpha_{i,j}^{Packet,Mode} = \frac{P_{i,j}^{Mode}}{N_{i,j}^{Mode}}, \tag{8}$$

where $\alpha_{i,j}^{Packet,Mode}$ are the ratios of the number of packets to the number of particles for experimental run $i$ and particle-size 280 class $j$ for the transport mode of saltation, rolling and sliding; $P_{i,j}^{Mode}$ and $N_{i,j}^{Mode}$ are the numbers of packets and transported particles for experimental run $i$ and particle-size class $j$ for the mode of saltation, rolling and sliding, respectively.

### 2.4.3 Estimation of particle velocity

Generally, the value of bedload particle velocity $V_P$ is expected to be less than the depth-averaged water flow velocity $V_W$. If the ratio $r_{PW} = V_P/V_W$ and $V_W$ are given, then $V_P$ can be estimated by the following equation:

$$V_P^{Est} = r_{PW}V_W, \tag{9}$$

where $V_P^{Est}$ is called the estimated particle velocity in present study; $r_{PW}$ ranges from 0.3 to 0.8 for natural particles as suggested by Julien and Bounvilay (2013).

$V_P$ can be also calculated by particle travel distance $L_P$ and time $\Delta T_P$, which is expressed as:

$$V_P^{Cal} = \frac{L_P^{SPG,MPA}}{\Delta T_P^{SPG,MPA}}, \tag{10}$$

where $V_P^{Cal}$ is called calculated particle velocity in this study; $L_P^{SPG,MPA}$ is a constant of 0.775 m, determined by the centre-to-centre distance between the SPG and MPA systems; $\Delta T_P^{SPG,MPA} = T_P^{MPA} - T_P^{SPG}$ is the arrival time difference determined from

the starting time of the packets $T_P^{SPG}$ and $T_P^{MPA}$ for the SPG and MPA systems, respectively. More details about the calculation of $\Delta T_P^{SPG,MPA}$ can be found in **Appendix C**.

To compare our experiments data with the other flume studies, the particle velocities introduced above are normalized as:

$$V_P^{M,*} = \frac{V_P^M}{\sqrt{(s-1)gD}}, \tag{11}$$

where $V_P^{M,*}$ represents the nondimensional particle velocity, i.e. $V_P^{Est,*}$ or $V_P^{Cal,*}$, indicating the particle velocity normalized by the term $\sqrt{(s-1)gD}$; $s$ is the ratio of particle density $\rho_s$ to water density $\rho$.

## 3 Results

### 3.1 Percentage distribution of transport modes

To assess signal signatures of the SPG system impacted by bedload particles varying in transport mode, a total of 2414 bedload impacts were analyzed for particles ranging in size (b-axis) from 28.1 mm to 171.5 mm (size classes C4 to C10) under a constant flow condition. **Fig. 6a** shows the total number of impacts for each bedload grain size class (identified in the video analysis), including the real impacts for the modes of saltation, rolling and sliding. The apparent impacts (bedload impacts on a neighboring plate or on the concrete bed) and the impacts that generate no packets are both included in the category "other impacts". Out of the total number of impacts, the percentage of the number of real impacts (= real packets, indicating packets triggered by bedload particles impacting on the plate above the considered geophone sensor) ranges from 22% to 31%, generally increasing with increasing particle size. As a consequence, the total number of real impacts over all particle sizes is calculated to be 571. The number of total recorded impacts equals to the number of real impacts and apparent impacts for all transport modes, considering all impact locations including the concrete bed and the SPG plates, as seen in **Fig. 6b**.

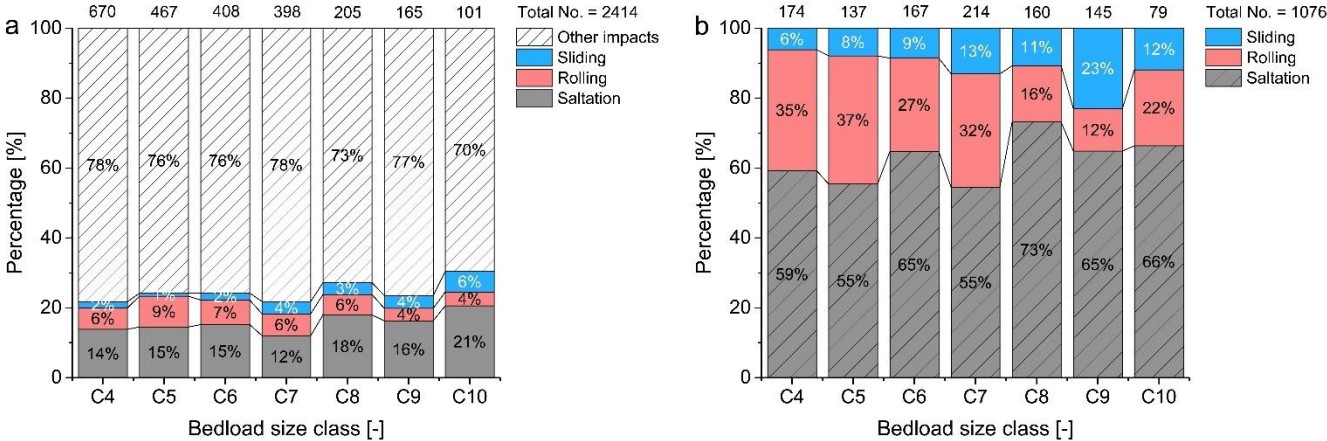

**Figure 6:** Percentage of impact numbers for the transport mode of saltation, rolling and sliding. (a) The total impacts (given at the top of each column) and the percentage of real impacts on the geophone plates for each transport mode. (b) The percentage

of total recorded impacts for all transport modes anywhere on the bed or plates. The number of total recorded impacts (given at the top of each column) indicates the sum of real impacts and apparent impacts, in which the real impacts correspond to the bedload impacts on the plate above the considered geophone sensor, while the apparent impacts represent the bedload impacts on a neighboring plate or on the concrete bed. The transport mode was determined through video analysis.

The value of $r_{i,j}{}^{Packet,V\_F}$ is slightly smaller than but close to one for small particle sizes ranging from 28.1 mm to 71.3 mm, (**Fig. 7**), indicating that the number of real packets based on the video analysis is smaller than that obtained from the filtering method using the amplitude-frequency information as introduced above. However, for the largest three particle size classes, the value of $r_{i,j}{}^{Packet,V\_F}$ generally increases with increasing particle size.

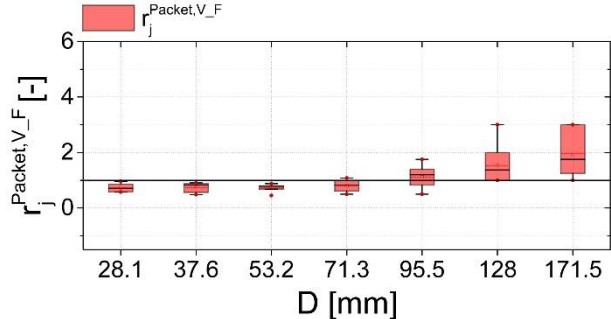

**Figure 7:** The ratio $r_{i,j}{}^{Packet,V\_F}$ of total number of real packets for all transport modes based on the video analysis to the number of real packets determined by the filtering method for each particle size class $j$.

### 3.2 Signal responses of the SPG system

In the following, we present the summary statistics of the coefficient $k_{IPM}$ (number of signal impulses extracted from the real packets per particle mass), the maximum amplitude $Amp_{Max,Pac}$ and the centroid frequency $Freq_{Centroid}$ as a function particle size or impact angle, respectively, for different transport modes. A constant impact velocity of 3.5 m/s for the spheres was used in the FEM simulations, resulting in different vertical impact velocities for different impact angles onto the plate (**Tab. 4**). The results of the inclined chute experiments are given in **Tab.5**.

**Table 5:** Results of the inclined chute experiments, including impact slope angles of 45° and 60°. Characteristic values of the number of signal impulses per particle mass $k_{IPM}$, the maximum amplitude $Amp_{Max,Pac}$ and the centroid frequency $Freq_{Centroid}$ were obtained from the SPG signal. The diameters of the spheres and b-axis length of the natural particles were 95.5 mm. The value range of $k_{IPM}$, $Amp_{Max,Pac}$, and $Freq_{Centroid}$ is given in brackets as $(25\%, 75\%)$ percentile.

| Particle type | $k_{IPM}$ [kg$^{-1}$] | | $Amp_{Max,Pac}$ [V] | | $Freq_{Centroid}$ [Hz] | |
|---|---|---|---|---|---|---|
| | 45° | 60° | 45° | 60° | 45° | 60° |
| Spherical | (27.6, 28.9) | (23.7, 27.6) | (7.4, 7.6) | (1.8, 8.5) | (1709.2, 1724.9) | (1671.2, 1675.7) |
| Natural | (13.6, 16.0) | (10.4, 18.4) | (1.8, 5.5) | (1.5, 6.0) | (1333.4, 1648.6) | (1249.9, 1505.3) |

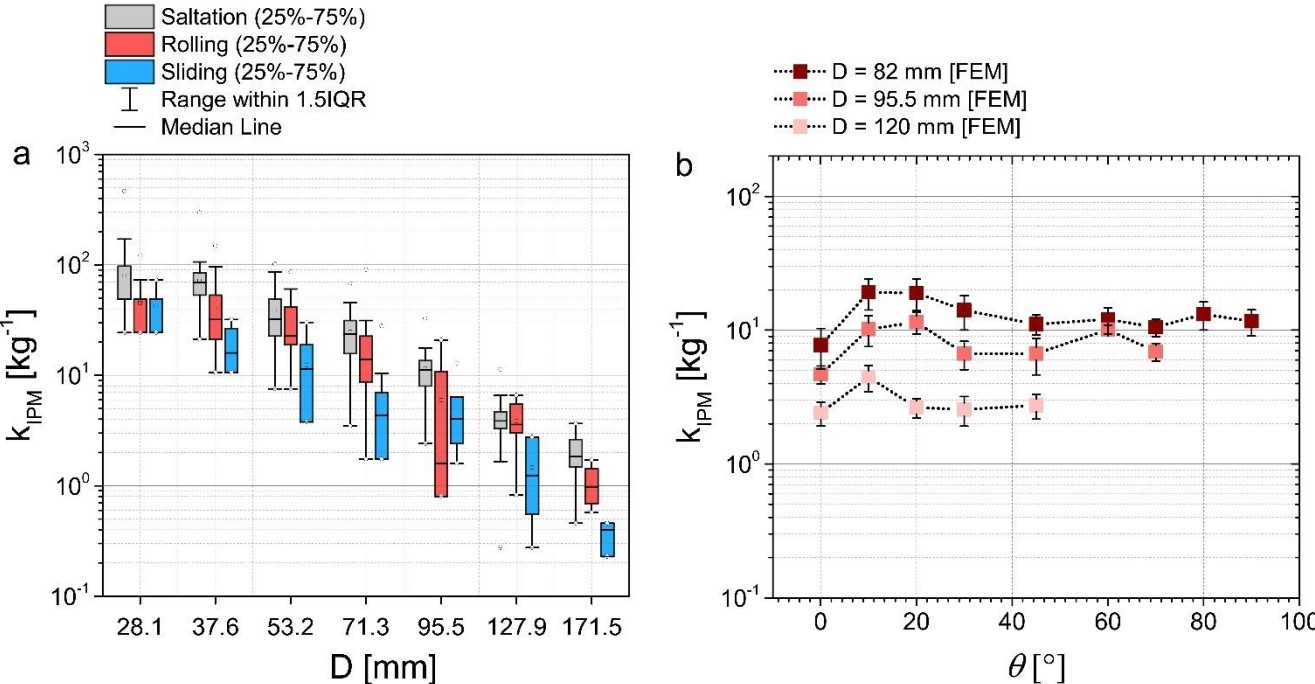

**Figure 8:** (a) Impulse-mass coefficient $k_{IPM}$ versus bedload particle size $D$ (b-axis) for different transport modes. (b) Impulse-mass coefficient $k_{IPM}$ versus the impact angle $\theta$ for different impacting particle sizes. FEM denotes simulations with the finite element method.

The coefficient $k_{IPM}$ decays strongly with increasing particle size $D$, regardless of whether the particles are in saltation, rolling or sliding motion (**Fig. 8a**). On average the $k_{IPM}$ values of saltation particles are larger than those of rolling particles, and the sliding particles tend to have the lowest values. The overlap of the $k_{IPM}$ values for particles in different transport modes varies between particle sizes which makes it difficult to distinguish motion modes by only considering the value of $k_{IPM}$.

According to the inclined chute experiments, the 75% percentile values of the impulse-mass coefficient $k_{IPM}$ change slightly with increasing slope angle ranging from 45° to 60°. However, compared to spheres, natural particles show a greater variation (25% to 75% percentile) in $k_{IPM}$ (**Tab. 5**). The FEM simulations indicate that $k_{IPM}$ varies only moderately with impact angle for a given particle size, except for impact angles changing from 0° to 10° for the FEM model (**Fig. 8b**). In contrast, the coefficients $k_{IPM}$ decrease with increasing sphere size. This is in an agreement with results from the flume experiments with natural particles (**Fig. 8a**).

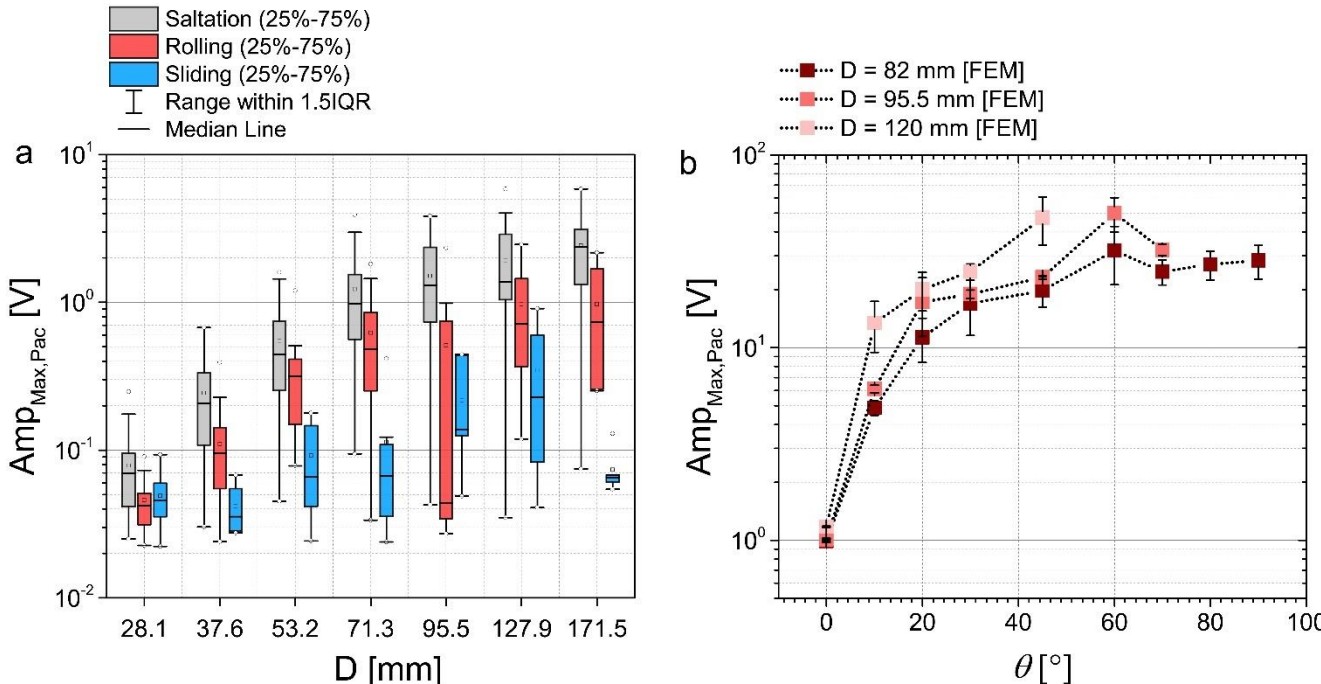

**Figure 9:** (a) Maximum amplitude $Amp_{Max,Pac}$ versus bedload particle size $D$ for different transport modes. (b) $Amp_{Max,Pac}$ versus impact angle $\theta$ for different particle sizes. FEM denotes simulations with the finite element method.

The maximum amplitude of a packet $Amp_{Max,Pac}$ generally increases with increasing bedload particle size $D$ for all transport modes (**Fig. 9a**). The saltation particles tend to generate the largest signal amplitudes, followed by the rolling particles and then the sliding particles. The sliding particles do not display a very clear relation between $Amp_{Max,\ Pac}$ and $D$.

The FEM simulations show that the maximum amplitude of a packet $Amp_{Max,Pac}$ increases with increasing particle impact angle $\theta$ up to about $\theta = 60°$ (**Fig. 9b**). The inclined chute experiments indicate better results for sphere impacts than that for natural particle impacts, and show a slightly increasing trend for the 75% percentile data due to limited range of slope angle (**Tab. 5**).

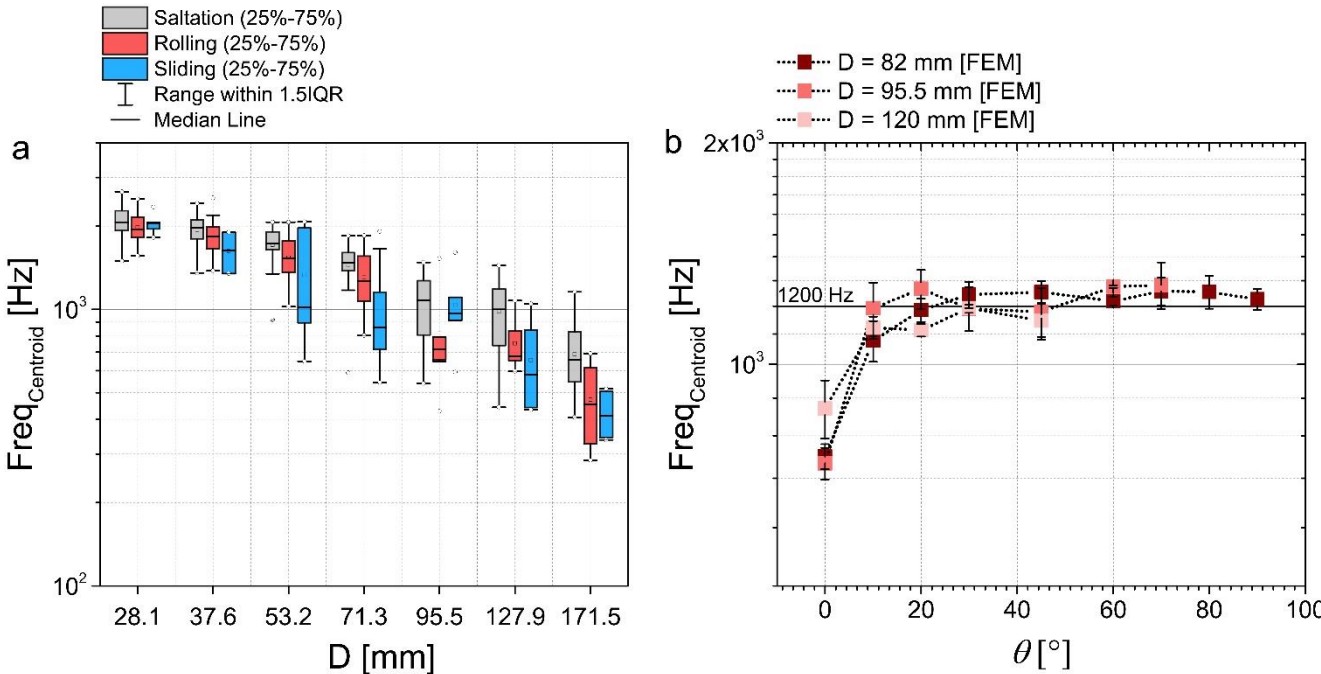

**Figure 10:** (a) Centroid frequency $Freq_{Centroid}$ versus bedload particle size $D$ for different motion modes. (b) Centroid frequency $Freq_{Centroid}$ versus impact angle $\theta$ for different impacting particle sizes. FEM denotes simulations with the finite element method.

The centroid frequency $Freq_{Centroid}$ generally decreases with increasing $D$ for all transport modes (**Fig. 10a**). Similar to the maximum amplitude, $Freq_{Centroid}$ values for saltation particles are generally largest, followed by values for the rolling and then the sliding particles. However, it appears that the discriminating effect of particle transport mode on the centroid frequency is rather weak for some particle sizes. The variability in frequency for each transport mode may also be partly due to variable particle impact locations on the geophone plate. Other factors, such as the particle shape can also play a role.

According to the FEM simulations, the centroid frequency $Freq_{Centroid}$ increases with impact angle up to about $\theta = 20°$ (**Fig. 10b**). The data from the inclined chute experiments show a slight decrease of $Freq_{Centroid}$ for the two tested impact angles (**Tab. 5**).

While there are discrepancies between the chute experiment data and the FEM results, the limited change of the characteristic values of the chute experiments with changing slope angle are in qualitative agreement with the FEM results with approximately constant characteristic values over a much larger range of slope angles from 20° to 90°.

## 4 Discussion

### 4.1 Effect of transport mode on the SPG signal response

#### 4.1.1 Number of packets for each transport mode

We showed that the ratio between the total number of real packets based on the video analysis and the number of packets resulting from the filtering method (Nicollier et al., 2021b, in review) $r_{i,j}{}^{Packet,V\_F}$ is slightly smaller than, but close to one for particle sizes ranging from 28.1 mm to 71.3 mm (**Fig. 7**). The ratio $r_{i,j}{}^{Packet,V\_F}$ is close to one for two reasons. First, in the video analysis, we considered only the signal packets that were generated by particle impacts on the SPG plates. Second, given that impacts of such small particles are generally too weak to generate apparent packets, the number of detected packets can be

expected to be close to the number of impacts on the SPG plates. A possible explanation for $r_{i,j}{}^{Packet,V\_F} < 1$ could therefore be the limited visibility during the video analysis due to flow turbulence, resulting in an underestimation of the number of impacts on the SPG plates. However, the value of $r_{i,j}{}^{Packet,V\_F}$ increases with increasing particle size ranging from 95.5 mm to 171.5 mm, and approaches a value of around two for the largest particle size class, which is possibly because of the following serval reasons: a) Some particles that impact close to boundaries (e.g. bolts) of the geophone plates were filtered out. b) The number

of impacts caused by sliding particles increases with increasing particle size. However, due to weak impact amplitude/energy, some sliding particles may be incorrectly filtered out. c) It has been noticed that for the largest size classes some bedload particles might be misclassified due to the filtering method itself. Nevertheless, the data from video analysis is in general agreement with that obtained from the filtering method (Nicollier et al., 2021b, in review).

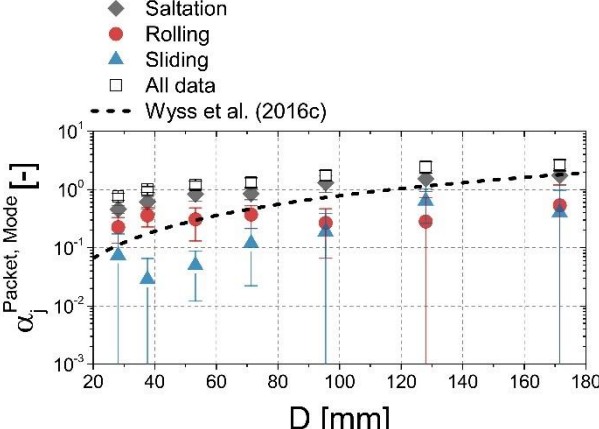

**Figure 11:** The number of packets divided by the number of transported particles for each transport mode ($\alpha_j{}^{Packet,Mode}$) as a function of bedload particle size $D$ ranging from 28.1 mm to 171.5 mm. "All data" represents the sum of the packets generated by saltation, rolling and sliding particles. The results from the flume experiments are also compared to the data of the Erlenbach field measurements analyzed by Wyss et al. (2016a) for a mean flow velocity of 5 m/s.

    The ratio $\alpha_j{}^{Packet,Mode}$ represents the number of packets (identified from the SPG signal) divided by the number of particles

transported over the plates (Wyss et al., 2016c). This represents a detection probability and can be considered as a calibration

curve of the SPG system, providing the values of $\alpha_j^{Packet,Mode}$ as a function of bedload particle size for each transport mode (**Fig. 11**). For the transport mode of saltation, the larger particles generally generate more packets recorded by the SPG system due to the higher impact energy leading to the longer wave transport distance. The values of $\alpha_j^{Packet,Mode}$ of the rolling and sliding particles change less with increasing particle size (**Fig. 11**), and they are relatively smaller than the values for the saltation

particles over all bedload particle size classes. This is likely due to the fact that, in this study, the bed shear stress is a constant during the flume experiments and is considerably larger than the critical bed shear stress, leading to a dominant transport mode of saltation.

The results obtained from the field measurements at the Erlenbach (Wyss et al., 2016) were compared with the results of our controlled flume experiments, showing relatively smaller values than the overall data and the saltation data based on the

video analysis. However, the differences between the field measurements data and the overall packets data of the flume experiments decrease with increasing particle size. It indicates that for the transport conditions in the Erlenbach, saltation appears to the dominant mode for $D >$ ca. 90 mm, while for $D <$ ca. 90 mm, the larger flow velocity at the Erlenbach could be the reason for less signal response there as compared to the Obernach flume data. Note that the field measurements conducted by Wyss et al. (2016) were associated with a mean water flow velocity of 5 m/s which is higher than the 3.3 m/s in our flume

experiments. As a consequence, the hop distance of a bedload particle in flow direction should be considerably larger for the field measurements than for the flume experiments, making it more likely for particles to fly over the plates, therefore, leading to the relatively smaller value of $\alpha_j^{Packet,Mode}$.

### 4.1.2 Impulses per particle mass

The impulse-mass coefficient $k_{IPM}$ decreases differently with increasing particle size for the different transport modes of

saltation, rolling and sliding. Generally, more impulses are triggered by the mode of saltation regardless of bedload particle size (**Fig. 8a**). This is possibly because the saltation particles have relatively higher hop heights and vertical impact velocities compared to the modes of rolling and sliding, under the same flow condition. The differences in $k_{IPM}$ between the rolling and sliding become significant with larger bedload particle sizes. This could be due to the following reasons: (i) For the packets data that were used to calculate $k_{IPM}$ it must be noted that the impact locations of bedload particles are variable, leading to the

differences between the rolling and sliding particles. (ii) The shape of large sliding particles is flatter than of the rolling particles, which may also contribute differently to the signal impulses.

### 4.1.3 Maximum amplitude

The maximum amplitude of a packet $Amp_{Max,Pac}$ is growing in nearly a power law form with increasing particle size for all the transport modes, especially for the modes of saltation and rolling (**Fig. 9a**). However, the values of $Amp_{Max,Pac}$ increase less

with changing bedload size for the largest particle sizes ranging from 127.9 mm to 171.5 mm, showing a qualitative agreement with the experiments at the Erlenbach (Wyss et al., 2016a) and with the FEM simulation data (Chen et al., 2021). The reason for this "saturation" limit in terms of maximum amplitude of a packet is likely due to a mechanical behavior of the SPG system.

The variation of signal amplitude for each particle size class and each transport mode is mainly considered to be caused by particle impact location on the SPG plates because of flowing water. Experimental results determined from laboratory drop tests and numerical data obtained from FEM simulations showed that the maximum amplitude was reduced by more than 50% with changes from centric impacts to the eccentric impacts (Chen et al., 2021). Note that even within a given particle size class in the flume experiments, the particles have a variable natural shape, which could also cause variable signal responses.

The median value of $Amp_{Max,Pac}$ for the mode of saltation is larger than that for the rolling and significantly larger than for the sliding particles (**Fig. 9**). This is because a particle in saltation generally has a higher impact velocity and can transfer more impact energy to the SPG plate. A considerable difference of $Amp_{Max,Pac}$ between the transport modes could potentially be helpful in identifying sliding particles for a given size experiment and therefore may improve the signal conversion into fractional bedload transport rates. However, the transport mode cannot be precisely identified using only the signal amplitude in natural field conditions. This leads to some challenges to further improve the particle-size identification by removing the effect of transport mode on the signal responses of the SPG system, because that the signal amplitude shows dependency on both particle size and on transport mode.

### 4.1.4 Centroid frequency

The frequency $Freq_{Centroid}$ of a generated signal decreases with increasing bedload particle size (**Fig. 10a**), showing an agreement with previous investigations (Rickenmann, 2017; Wyss et al., 2016a). The median value of $Freq_{Centroid}$ for saltation particles is slightly larger than that for rolling and sliding particles. Assuming that the vertical impact velocity generally decreases from saltation to rolling to sliding particles, the observed change in $Freq_{Centroid}$ with changing transport mode is in general agreement with the Hertz theory (Thorne, 1985). According to the Hertz contact theory (Johnson, 1985; Thorne, 1986), the characteristic frequency of the signal response of the geophone plate shows a dependency with the contacting particle size (Bogen and Møen, 2003; Barrière et al., 2015; Wyss et al., 2016b; Rickenmann, 2017), indicating that the frequency decreases with increasing particle size. However, for a given particle size class, the differences of $Freq_{Centroid}$ between the three transport modes are not very significant. In any case, among all contributing factors, particle size dominates the centroid frequency according to the Hertz theory. Although all signal data were obtained under the same constant flow conditions, the velocity of saltation particles is relatively larger but not considerably larger than that in rolling and sliding. The rolling and sliding bedload particles with approximately the same impact velocity move near the flume bed, resulting in little difference in frequency for the same size class.

### 4.2 Effect of particle impact angle on the SPG signal response

The impact angle $\theta$ between the directions of the water flow and the bedload particle motion might have an influence on the signal responses of the SPG system because of the changes of the vertical and horizontal components of impact velocity. The impulse-mass coefficient $k_{IPM}$ changes only moderately with increasing impact angle ranging from around 5° to 90° as seen in numerical results (**Fig. 8b**), which were compared with the inclined chute experiments with both spherical and natural particles

for the impact angles of 45° and 60°. However, a clear effect of bedload particle size on $k_{IPM}$ can be observed in **Fig. 8b**, indicating that the value of $k_{IPM}$ is reduced with increasing particle size, which shows a reasonable agreement with previous findings (Chen et al., 2021). This means that the SPG monitoring system is more sensitive to the bedload particle size than to the impact angle, in agreement with the Hertz theory as indicated above.

The maximum amplitude $Amp_{Max,Pac}$ increases with increasing impact angle for the numerical data up to an intermediate angle of about 45° (**Fig. 9b**). The values of $Amp_{Max,Pac}$ for the FEM simulations are considerably larger than those from the chute experiments for the impact angles of 45° and 60°. This may be partly because that the impact velocities in the inclined chute experiments were overestimated. Note that the impact velocities calculated from the experimental videos were variable even for a fixed release height and particle size, due to friction along the chute bed and drag forces of the water. The curves of $Amp_{Max,Pac}$ to impact angle tend to become flatter with increasing impact angle. This is possibly due to a plastic behavior of the plate material, as the vertical velocity component becomes relatively large compared to typical natural flow conditions.

For a given impacting sphere size, the centroid frequency $Freq_{Centroid}$ appears to be relatively insensitive to changing impact angle except for nearly horizontal impacts (**Fig. 10b**). $Freq_{Centroid}$ is comparatively lower for impact angles ranging from 0° to 10° than for the rest of impact angles, which can be possibly explained as follows. We consider the fact that the horizontal impacts (sliding mode) in the FEM simulations are under a perfect condition with an impacting angle of 0°, indicating that a contact between the spherical particle and the SPG plate is dominated by friction. It's convenient to assume that as the impact angle approaches horizontal, the normal stress goes down while the shear stress increases. Furthermore, $Freq_{Centroid}$ can drop due to the extremely low vertical impact velocity (see **Tab. 4**) for the horizontal impact. As a consequence, the signal response and wave propagation could be fundamentally different with the circumstances of non-horizontal impacts, leading to a lower signal centroid frequency.

The centroid frequency $Freq_{Centroid}$ has been found to be somewhat less sensitive to varying flow velocities than the maximum packet amplitude $Amp_{Max,Pac}$, based on controlled flume experiments (Wyss et al. 2016). In the present study, we found that the dependences of $Freq_{Centroid}$ on transport mode (as stated previously) and impact angle are less than that of $Amp_{Max,Pac}$ on transport mode and impact angle. Therefore, the centroid frequency appears to be somewhat better suited for particle size identification than the maximum amplitude.

**4.3 Comparison with other flume studies**

**4.3.1 Probability of transport mode**

The probability of occurrence of each transport mode is related to the flow intensity or the transport stage (Auel et al., 2017a; Hu and Hui, 1996a), indicating correlations with the bedload size as well. Note that the transport mode of rolling and sliding are not distinguished in Auel et al. (2017a), both modes are lumped together in rolling mode. In the flume experiments

conducted by Auel et al. (2017a), sediment particles of three size categories, namely small, medium and large, ranging from about 5.3 mm to 17.5 mm were investigated in an artificial channel and recorded using the high-speed camera. Subsequently,

the regression line that represents a shift from the saltation mode to the rolling mode was obtained, considering partial data from Hu and Hui (1996a), as seen in **Fig. 12a**. Auel et al. (2017a) defined the probability for the rolling mode as the ratio of the travelled distance covered by a rolling particle to the overall travelled distances determined by the sum of all transport modes that are averaged over numbers of particles. Comparably, in our study, the probability is calculated as the number of signal packets generated by particle impacts for each transport mode divided by the total recorded packets. Since that the number of total recorded packets can represent the number of particles that were transported over the geophone plates and the surrounding concrete, the definition of our study is more similar to that which was used in Hu and Hui (1996a).

The probability of rolling mode $P_{Rol}$ decreases in a power law form with increasing excess transport stage $T$ for the data compiled by Auel et al. (2017a), as is also illustrated with their proposed power law model to distinguish between the rolling and saltation regions in **Fig. 12a**. In our study, the dominant transport mode is saltation, for $P_{Sal}$ equals about 55% to 73% varying in $T$. In contrast, 12% to 37% of the particles are in rolling mode, followed by the sliding mode that shows about 6% to 23%. We included our experimental data in **Fig. 12a** by defining the cumulative probabilities $P_{Sli} + P_{Rol} + P_{Sal} = 1$, based on the data from **Fig. 6b**. The changes in the probabilities of each transport mode are due to different particle sizes $D$ ranging from 28.1 mm to 171.5 mm that leads to values of $T$ changing from 6.42 to 0.22. With regard to $P_{Rol}$, the results of the flume experiments in Auel et al. (2017a) indicated that large particles have a high probability $P_{Rol}$ for similar $T$ values as in our study. For the three smallest $T$ values our data show that the sum $(P_{Rol} + P_{Sli})$ values are somewhat smaller whereas $P_{Sli}$ is slightly larger than for other (higher) $T$ values. For small $T$ values the bed shear stress is very close to that of incipient motion of particles, and for more angular or flatter-shaped particles this might have caused a decrease in the $P_{Rol}$ values. Indeed, flatter-shaped particles are more likely to move in the sliding mode according to our video observations. For the four largest $T$ values, the rolling and saltation particles of our experimental data are reasonably consistent with the data of Auel et al. (2017a).

### 4.3.2 Particle velocity

Our experimental data for the dimensionless particle velocity $V_P^*$ show a dependency on the transport stage $T$, indicating a power law (**Fig. 12b**). A similar trend was found by Auel et al. (2017a), who compared their data with experimental data from other studies (Abbott and Francis, 1997; Ancey et al., 2008; Chatanantavet, 2007; Chatanantavet et al., 2013; Fernandez Luque and Van Beek, 1976; Hu and Hui, 1996; Ishibashi and Isobe, 1968; Lee and Hsu, 1994; Lajeunesse et al., 2010; Sekine and Kikkawa, 1992). $V_P^*$ represents the particle velocity $V_P$ ($V_P^{Cal,*}$ and $V_P^{Est,*}$) normalized by the term $\sqrt{(s-1)gD}$, where $V_P^{Cal,*}$ is the particle velocity calculated by particle travel distance and arrival time difference determined from the starting time of the packets of the SPG and MPA signals; $V_P^{Est,*}$ is the estimated particle velocity, assuming that the ratio (30% to 80%, the red shaded area in **Fig. 12b**) of particle velocity to the flow velocity is known.

Our results indicate that the estimated particle velocity is in the range of about 53% to 88% of the flow velocity, showing that it is slightly higher than but basically agrees with the range (the transparent red shaded area in **Fig. 12b**) given in Julien and Bounvilay (2013). The data are also close to the empirical model presented by Auel et al. (2017a), with the largest deviation

for the lowest *T*. The disagreement between our flume experiments and the power law function (see red dotted line in **Fig.12b**) of Auel et al. (2017a) is most pronounced for the smallest value of *T*. A reason might be that the bed stress was particularly close to that at incipient particle motion, and for such conditions there is generally a larger scatter of all experimental data than for larger values of *T*. Nevertheless, the general agreement of most of our experimental data on particle velocities suggests that our observations on particle transport modes should also be comparable with other flume studies.

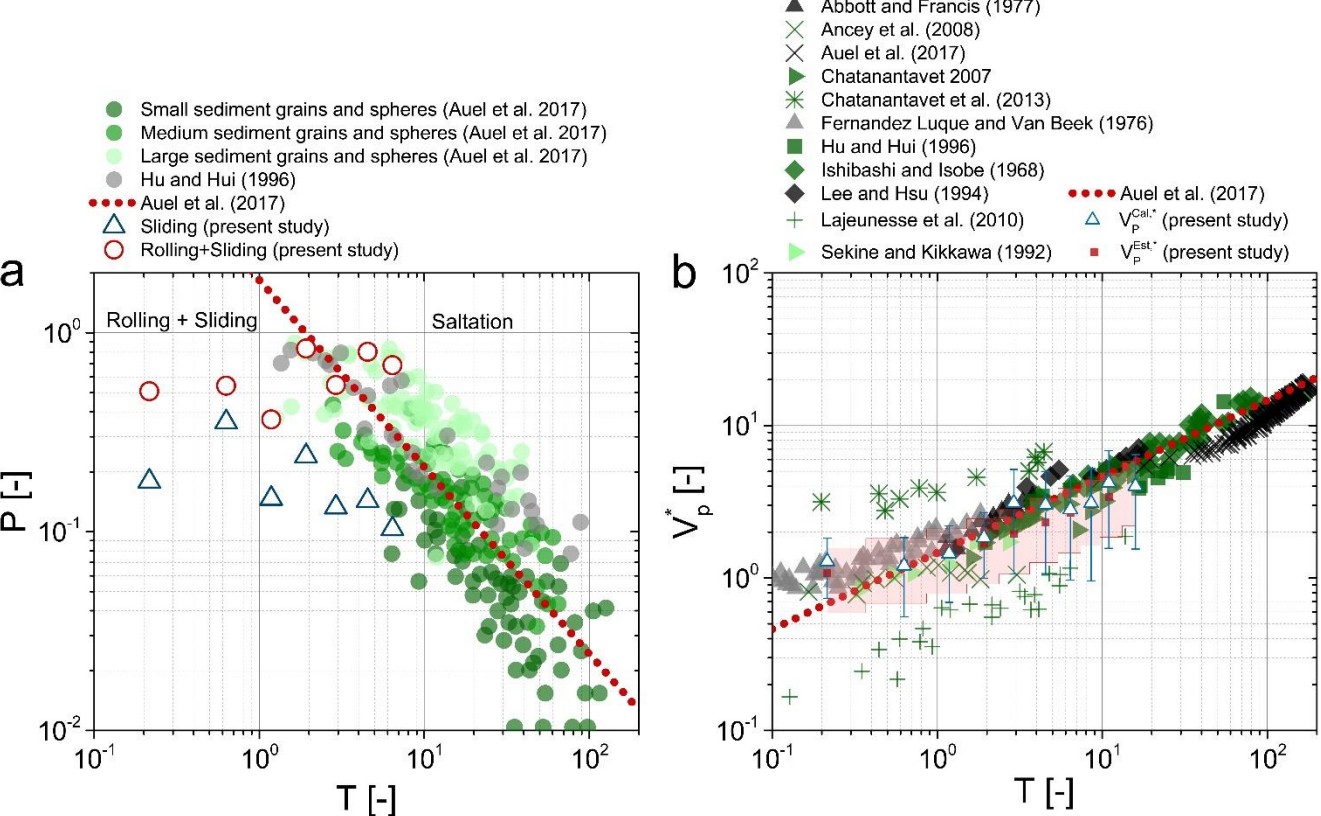

**Figure 12:** (a) Probability of transport mode $P$ ($P_{Sal}$ for saltation, $P_{Rol}$ for rolling, and $P_{Sli}$ for sliding) as functions of the excess transport stage $T$. The data from Auel et al. (2017a), associated with natural grains and spheres with variable sizes ranging from about 5.3 mm to 17.5 mm, are shown in green. The data from Hu and Hui (1996a) are presented in gray. The present results from our flume experiments in terms of $T$ ranging from 0.22 to 6.42 are compared to the power law model $P = 1.84T^{-0.94}$ applied by Auel et al. (2017a) (red dotted line), with $R^2 = 0.64$, and their compiled data. (b) The non-dimensional particle velocity $V_P^* = V_P / \sqrt{(s-1)gD}$ versus $T$. The power law model $V_P^* = 1.46T^{0.5}$ applied by Auel et al. (2017a) is shown in red dotted line, with $R^2 = 0.95$. The transparent red shaded area indicates variabilities of the particle velocity $V_P^{Est,*}$ estimated from the flow velocity, ranging from 30% to 80% as suggested by Julien and Bounvilay (2013).

## 5 Conclusions

In this research, systematic flume experiments and FEM simulations were conducted to study the signal response of the Swiss plate geophone bedload monitoring system when impacted by natural bedload particles varying in size, and showing different

angles of impact and transport modes. Some key parameters of the acoustic signal have been analyzed, including the ratio of the number of packets to the number of transported particles $\alpha_j^{Packet,Mode}$, the maximum amplitude of a packet $Amp_{Max,Pac}$, the impulse-mass coefficient $k_{IPM}$, and the centroid frequency $Freq_{Centroid}$. The major conclusions of this study are summarized as follows:

[1] The number of impacts counted from the experimental video is in general agreement with the data obtained from the filtering method. The number of packets for the rolling and sliding particles changes less with increasing particle size. Also, for all bedload particle size classes, sliding and rolling generate smaller number of packets than saltation.

[2] The number of signal impulses per unit particle mass decreases nonlinearly with increasing bedload particle size, and displays a dependency on particle transport mode. It only weakly depends on particle impact angle. In general, saltating 550 particles trigger a larger number of signal impulses than rolling and sliding particles.

[3] The maximum amplitude of a signal packet increases with increasing particle size for the saltating and rolling particles, showing a dependency on particle impact angle. The strongest signal response of the SPG system is excited by the saltation particles, followed by the rolling particles, and the weakest signal is triggered by the sliding particles.

[4] The centroid frequencies of the acoustic signal generally decrease with increasing particle size across all transport modes. 555 For the FEM simulations, the centroid frequency values are considerably lower for the horizontal impact than for the rest of impact angles for a given particle size, indicating differences between the sliding and the saltation particles. The centroid frequency appears to be somewhat better suited for particle size identification, as it is less sensitive to varying transport modes and impact angles, than the maximum packet amplitude. This finding could be helpful for further improving the analysis of the SPG signal for fractional transport estimation.

[5] The probability of each transport mode correlates with the transport stage and particle size of the bedload. The dominant transport mode in this study is saltation, and the non-dimensional velocity of bedload particle increases in a power law form with increasing transport stage, and is in general agreement with other flume studies.

**Appendix A**

Before the numerical simulations, the FEM model has been calibrated with results obtained from the previous lab experiments 565 (drop tests) with quartz spheres (see below, Chen et al. 2021). Systematic drop experiments (**Figs. A1a** and **A1b**) have been performed in quiescent water in the experimental flume with spheres of known mass and diameter. A transparent Plexiglas tube was used to control the drop height and to prevent any horizontal flow from disturbing the vertical fall trajectory of the sphere. The impact velocity of each quartz sphere falling onto the plate was measured in the laboratory tests (Gaillard, 2018) using a high-speed camera mounted with a side view. **Fig. A1c** shows a comparison of the FEM signal and the experimental 570 signal during the drop test in the Z-direction (perpendicular to the plate and pointing up), triggered by a single bedload particle (diameter $D = 120$ mm) impacting the plate (at centric location) with a velocity of 0.777 m/s.

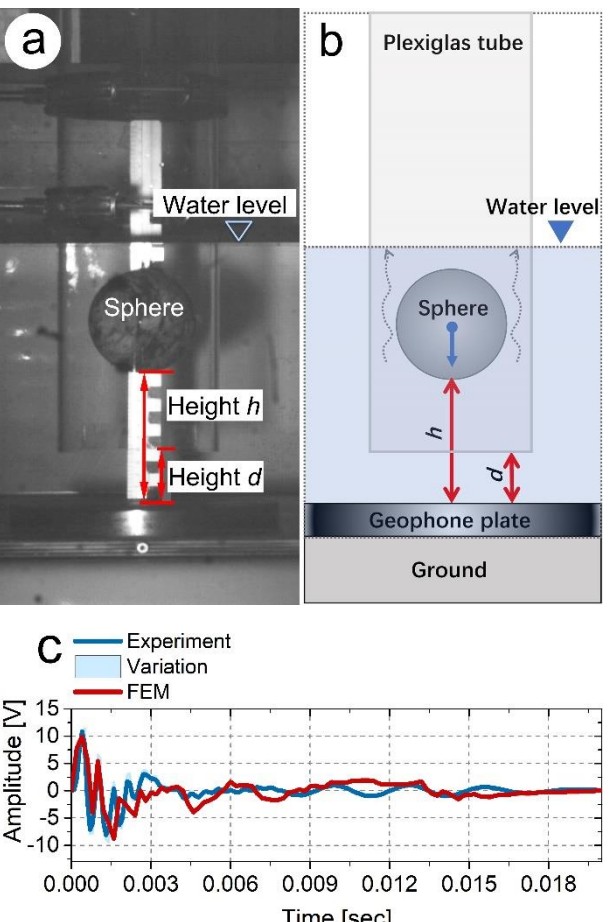

**Figure A1:** (a) Side-view photo (Gaillard, 2018) and (b) sketch (taken from Chen et al. 2021) of the drop-test set-up used in earlier laboratory tests to measure the impact velocity. $h$ is the distance between the bottom surface of the sphere and the plate, and $d$ is the distance between the bottom of the Plexiglas tube and the plate. The tube protects the sphere from flow turbulences in cases where the set-up is used at field sites. (c) Comparison of the FEM signal and the experimentally generated signal (from the drop test) in the Z-direction (perpendicular to the plate and pointing up), triggered by a single bedload particle (diameter $D = 120$ mm) impacting on the centric location of the plate with a velocity of 0.777 m/s.

## Appendix B

The analysis of the experimental videos included the following five steps: (1) tracking a saltating bedload particle from frame to frame during a certain time duration, especially when the particle contacts with the SPG plate or a nearby location, because saltation generally triggers a higher signal amplitude than the other two transport modes; (2) determining the time instants (time series $T_m^V$) of each impact caused by this particle from the video frames, including the observation of a slight rotation (at time $T_{m_0}^V = T_2$) of the particle at the contact point, as described above; (3) isolating the signal packet (at time $T_{i_0}^S$) from the SPG output signals, as this packet is indicative of the particle impact on the SPG plates; (4) matching the analyzed particle

impacts with the SPG signals, using the formula **Eq. B1** and satisfying the condition of **Eq. B2**; (5) checking the impact instants generated by the rolling and sliding particles.

$$T_i^{S,Cal} = \lambda(T_m^V - T_{m_0}^V) + T_{i_0}^S,$$ (B1)

$$\left|T_i^S - T_i^{S,Cal}\right| < 3 \times 10^{-3} \text{ sec},$$ (B2)

where $\lambda = 1/3$ is a coefficient for correcting the video time, $T_i^{S,Cal}$ is the calculated time instant for each signal packet, $T_m^V$ is the time instant of each bedload impact based on video observation, $T_{m_0}^V$ is the representative impact instant based on video observation, $T_{i_0}^S$ is the time instant for the isolated signal packet matched with $T_{m_0}^V$, and $T_i^S$ is the packets' time series recorded by geophones. The upper limit is considered in **Eq. B2** because, in general, the contact time between the particle and the plate

ranges from one to three milliseconds, which is less than the packet duration that typically lasts five to ten milliseconds.

**Appendix C**

**Figs. C1a** and **C1b** show representative signals of the SPG and MPA systems. The arrival time difference $\Delta T_P^{SPG,MPA}$ between the systems can be calculated from the starting time of the packets $T_P^{SPG}$ and $T_P^{MPA}$ for the SPG and MPA as $\Delta T_P^{SPG,MPA} = T_P^{MPA} - T_P^{SPG}$, noting that the MPA system is located at the downstream position of the SPG system Thus, the question now is

to determine $T_P^{SPG}$ and $T_P^{MPA}$.

Given a time window $T^W$ and time step $\Delta t_P$ (as seen in **Figs. C1a** and **C1b**), the number of packets $P^{W_k}$ within the $k^{th}$ time window $T_k^W$ can be counted as:

$$P^{W_k} = \sum_1^n m^{W_k},$$ (C1)

where $k = 1, 2, \ldots, \left\lceil \frac{T^E - T^W}{\Delta t_P} \right\rceil$; $T^E$ is time duration of an experiment; "$\lceil \quad \rceil$" is the ceiling operator; $n$ is the value of number of

packets $P^{W_k}$, varying with the moving time window; $m = 1$.

In our study, the time window and time step are given as 1.0 s and 0.05 s, respectively. Consequently, the number of packets for the SPG and MPA system over the experimental duration $T^E$ can be expressed as functions of time, corresponding to the blue and red lines in **Figs. C1c** and **C1d**, respectively. As the final number, we utilize the time difference that accounts for 5% of the maximum value, as seen below:

$$\Delta T_P^{SPG,MPA} = T_P^{MPA,5th} - T_P^{SPG,5th},$$ (C2)

where $T_P^{MPA,5th}$ and $T_P^{SPG,5th}$ correspond to 5% of the maximum value in **Fig. C1c**.

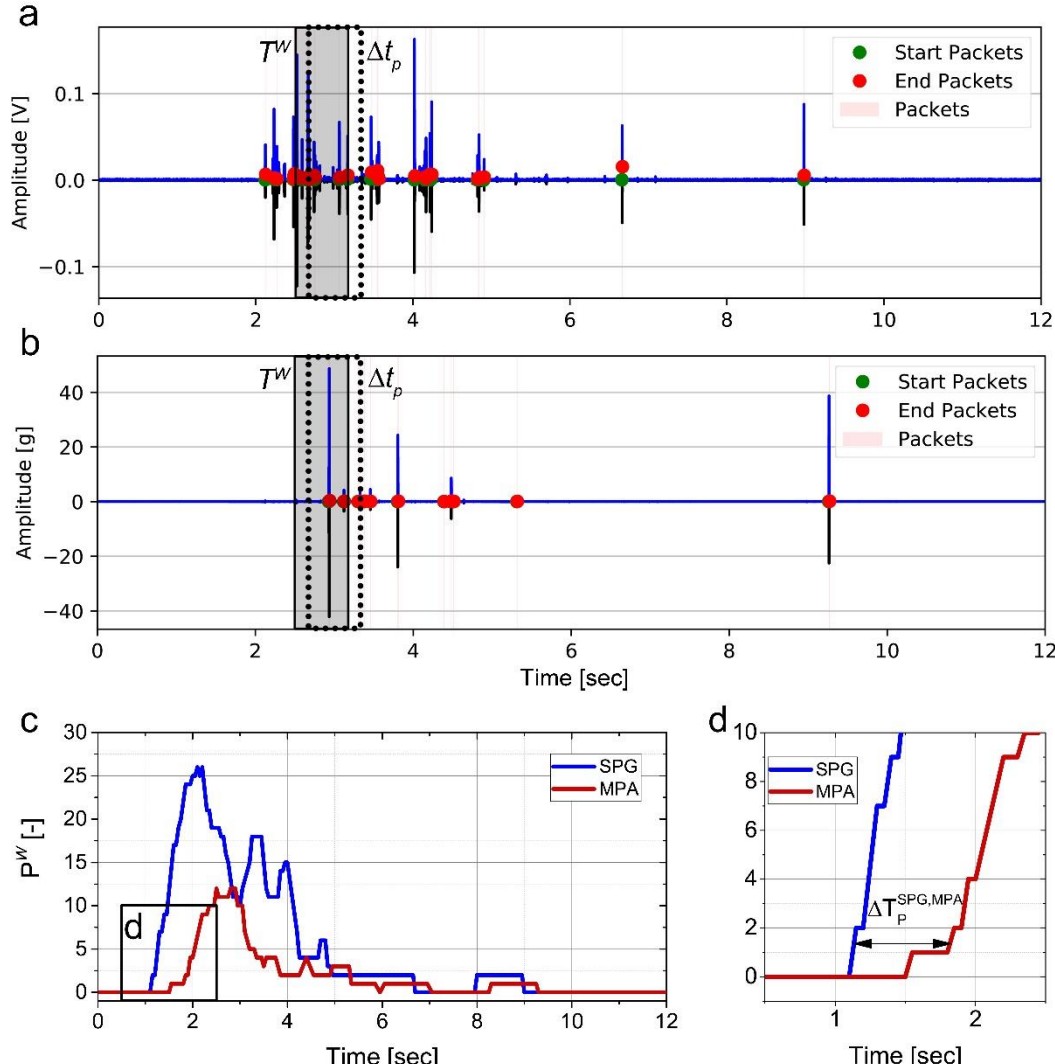

**Figure C1:** Illustration of the vibrations, signal packets and counts of packet number, following a flume experiment with bedload particles of grain-size class C4 and with a flow velocity of 3.3 m/s. (a) and (b) are representative signals that were recorded by the SPG and MPA systems, respectively. (c) and (d) Counting the number of packets, summing up the numbers within the given time window (gray shaded area in (a) and (b)). The blue and red lines are the summed number of packets of each time window for the SPG and MPA systems, respectively.

### Code availability statement

Please contact Dieter Rickenmann (dieter.rickenmann@wsl.ch) and Tobias Nicollier (tobias.nicollier@wsl.ch) if readers are interested in the code used in this paper.

**Data availability statement**

Datasets for this research are available upon request to the readers.

**Author contribution**

**Zheng Chen:** Formal analysis, Investigation, Methodology, Software, Visualization, Writing – original draft preparation, Writing – review & editing; **Siming He:** Software, Writing – review & editing; **Tobias Nicollier:** Data curation, Investigation, Methodology, Software, Writing – review & editing; **Lorenz Ammann:** Investigation, Writing – review & editing; **Alexandre Badoux:** Supervision, Writing – review & editing; **Dieter Rickenmann:** Conceptualization, Funding acquisition, Resources, Supervision, Validation, Writing – review & editing.

**Competing interest**

The authors declare that they have no conflict of interest.

**Funding information**

This study was supported by the National Natural Science Foundation of China (grants 41772312 and 41790433 awarded to SH), the Key Deployment Project of CAS (grant KFZD-SW-424 awarded to SH), the China Scholarship Council CSC (file no. 201904910867 awarded to ZC), and the Swiss National Science Foundation SNSF (grant 200021L_172606 awarded to DR).

**Acknowledgement**

The authors thank Arnd Hartlieb and his colleagues of TU Munich for their support with the experiments in the Obernach outdoor flume facility. The authors also warmly thank the associate editor Rebecca Hodge, the reviewer Thomas Geay and one anonymous reviewer for their valuable comments on this paper.

**Notations**

| Symbols | Descriptions |
| --- | --- |
| $T_{i_0}^S$ | time instant for an isolated signal packet [s] |
| $T_{m_0}^V$ | representative impact instant based on video observation [s] |
| $A_{FFT,\,m}$ | amplitude that is obtained by performing fast Fourier transform FFT on the signals [V·s] |

| | |
|---|---|
| $Amp_{Max,Pac}$ | maximum positive amplitude of the packet [V] |
| $b$ | channel width [m] |
| $D_i$ | mean value of bedload particle diameter for each size class $i$ [mm] |
| $\boldsymbol{F_c}$ | friction force exerted by the SPG plate on the bedload particle [N] |
| $F_{Contact}$ | contact force between the sphere and the plate [N] |
| $FEM$ | finite element method [-] |
| $FFT$ | fast Fourier transform [-] |
| $f_m$ | spectrum frequency [Hz] |
| $\boldsymbol{F_n}$ | vertical support force exerted by the SPG plate on the bedload particle [N] |
| $FPS$ | frames per second [s$^{-1}$] |
| $Freq_{Centroid}$ | centroid frequency of acoustic signals [Hz] |
| $\boldsymbol{F_w}$ | force of water acting on the bedload particle [N] |
| $\boldsymbol{G}$ | particle weight force [N] |
| $g$ | gravity acceleration [m s$^{-2}$] |
| $h$ | flow depth [m] |
| $I$ | number of impulses recorded by the SPG system for each impact event [-] |
| $I_j$ | impulses recorded by the SPG system for bedload particle-size class $j$ [-] |
| $JPM$ | the Japanese pipe microphone [-] |
| $k_{IPM}$ | number of impulses per particle mass that is transported [kg$^{-1}$] |
| $L_P$ | particle travel distance [m] |
| $L_P^{SPG,MPA}$ | centre-to-centre distance between the SPG and MPA systems [m] |
| $m$ | Number of tests [-] |
| $M$ | transported bedload mass [kg] |
| $M_j$ | mean value of bedload particle mass for each size class $j$ [kg] |
| $MPA$ | the miniplates accelerometer [-] |
| $n$ | number of particles for each experimental run [-] |
| $N_{i,j}$ | number of particles for each experimental run $i$ and grain-size class $j$ [-] |
| $N_{i,j}^{Mode}$ | number of particles for experimental run $i$ and particle-size class $j$ for the transport mode of saltation, rolling and sliding [-] |
| $N_{i,j}^{Packet,F}$ | number of real packets for experimental run $i$ and particle-size class $j$ determined by the filtering method [-] |
| $N_{i,j}^{Packet,V}$ | total number of real packets for experimental run $i$ and particle-size class $j$ for all transport modes based on the video analysis [-] |
| $PDE$ | partial differential equations [-] |
| $P^{W}{}_{k}$ | number of packets within the $k^{th}$ time window [-] |
| $P_{i,j}$ | number of packets for each experimental run $i$ and grain-size class $j$ [-] |
| $P_{i,j}^{Mode}$ | number of packets for experimental run $i$ and particle-size class $j$ for the motion mode of saltation, rolling and sliding [-] |
| $P_M$ | probability of transport mode ($P_{Sal}$, $P_{Rol}$, and $P_{Sli}$ for saltation, rolling, and sliding, respectively) [-] |
| $R_h$ | hydraulic radius [m] |

| | |
|---|---|
| $r_{i,j}^{Packet,V\_F}$ | ratio of the total number of real packets for all transport modes based on the video observations to the number of real packets for experimental run $i$ and particle-size class $j$ determined by numerical filtering method [-] |
| $r_{PW}$ | ratio of particle velocity to water flow velocity [-] |
| $S$ | bed slope [-] |
| $s$ | ratio of particle density to water density |
| $SPG$ | the Swiss plate geophone [-] |
| $T$ | excess transport stage [-] |
| $T^E$ | time duration of an experiment [-] |
| $T_i^S$ | packets' time series recorded by geophones [s] |
| $T_i^{S,Cal}$ | calculated time instant for each signal packet [s] |
| $T_k^W$ | The $k^{th}$ time window [s] |
| $T_m^V$ | time instant of each bedload impact based on video observation [s] |
| $T_P^{MPA}$ | starting time of the packets for the MPA system [s] |
| $T_P^{MPA,5th}$ | starting time of the packets for the SPG system, corresponding to 5% of the maximum value [s] |
| $T_P^{MPA,5th}$ | starting time of the packets for the MPA system corresponding to 5% of the maximum value [s] |
| $T_P^{SPG}$ | starting time of the packets for the SPG system [s] |
| $T^W$ | time window [s] |
| $V$ | impact velocity of the sphere onto the plate [m s$^{-1}$] |
| $V_P$ | particle velocity [m s$^{-1}$] |
| $V_P^*$ | dimensionless particle velocity [-] |
| $V_P^{Cal}$ | calculated particle velocity [m s$^{-1}$] |
| $V_P^{Cal,*}$ | particle velocity calculated by particle travel distance and time lag determined from the SPG and MPA signals [-] |
| $V_P^{Est}$ | estimated particle velocity [m s$^{-1}$] |
| $V_P^{Est,*}$ | particle velocity estimated from the ratio of the averaged particle velocity to water flow velocity [-] |
| $V_P^{M,*}$ | nondimensional particle velocity $V_P^{Est,*}$ or $V_P^{Cal,*}$ [-] |
| $V_W$ | water flow velocity [m s$^{-1}$] |
| $V_Y$ | Y-component of the impact velocity [m s$^{-1}$] |
| $V_Z$ | Z-component of the impact velocity [m s$^{-1}$] |
| $\alpha_{i,j}^{Packet}$ | ratio of the number of packets to the number of particles for each experimental run $i$ and grain-size class $j$ [-] |
| $\alpha_{i,j}^{Packet,Mode}$ | ratios of the number of packets to the number of particles for experimental run $i$ and particle-size class $j$ for the motion mode of saltation, rolling and sliding [-] |
| $\Delta T_P$ | particle travel time [s] |
| $\Delta t_P$ | time step [s] |
| $\Delta T_P^{SPG,MPA}$ | arrival time difference determined from the starting time of the packets and for the SPG and MPA systems [s] |
| $\theta$ | impact angle [°] |

| $\Theta_{Critical}$ | critical Shields parameter [-] |
| $\lambda$ | coefficient for correcting the video time [-] |
| $\rho$ | water density [kg m$^{-3}$] |
| $\rho_s$ | particle density [kg m$^{-3}$] |
| $\overline{\tau_b}$ | time-averaged bed shear stress [N m$^{-2}$] |
| $\tau_{critical}$ | critical bed shear stress [N m$^{-2}$] |

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
