# Peer review of "Signal response of the Swiss plate geophone monitoring system impacted by bedload particles with different transport modes"

_Earth Surface Dynamics, 2021_

## Referee Comment (RC2)

**%------------------ Introduction ------------------%**

**L39** . Could be valuable to add other papers using direct calibration. For example:

Bakker, M., Gimbert, F., Geay, T., Misset, C., Zanker, S., & Recking, A. (2020). Field Application and Validation of a Seismic Bedload Transport Model. Journal of Geophysical Research: Earth Surface, e2019JF005416. https://doi.org/10.1029/2019JF005416

Geay, T., Zanker, S., Misset, C., & Recking, A. (2020). Passive Acoustic Measurement of Bedload Transport: Toward a Global Calibration Curve? Journal of Geophysical Research: Earth Surface, 125(8), 1–19. https://doi.org/10.1029/2019JF005242

Rennie, C. D., Vericat, D., Williams, R. D., Brasington, J., & Hicks, M. (2017). Calibration of acoustic doppler current profiler apparent bedload velocity to bedload transport rate. Gravel-Bed Rivers: Process and Disasters, 209–233. https://doi.org/10.1002/9781118971437.ch8

**L56-58.** To my mind, Thorne [1985] has not shown that transport modes are associated to different acoustic response (it is only suggested in this paper). Please check my remark and remove this reference if this review is true.

**%------------------ Methods ------------------%**

**Table 1 -** When monitoring bedload, we are interested in estimating bedload fluxes (in kg/min, g/m/s or else). In the presentation of the Flume experiments (section 1.1.2), I think important to have an estimation of these bedload fluxes (even if imprecise). I believe that the number of moving particles strongly affects the modes of transport and their associated characteristics (impact velocities/angles, etc.), due to interaction between particles. This was suggested in the following paper: Gimbert, F., Fuller, B. M., Lamb, M. P., Tsai, V. C., & Johnson, J. P. L. (2018). Particle transport mechanics and induced seismic noise in steep flume experiments with accelerometer-embedded tracers. Earth Surface Processes and Landforms. https://doi.org/10.1002/esp.4495. The number of moving particles is dependent on bedload fluxes and bedload sizes. For me, it is therefore very important to give an overview of the bedload fluxes used in these experiments.

**L128-130** - What are the average impact velocities of the particles in the flow (i.e. in the flume experiments)? Is it comparable to the particle velocity in the inclined chute experiments? Please add a sentence on this comparison.

**Figure 2b.** Add "view" to read "Cross-sectional view of the FEM model of the SPG system."

**L154** - Maybe delete "the" from "the bedload particles".

**L210** –is $k_{IPM}$ comparable to the $k_b$ of almost all other studies [Rickenmann et al., 2013; Wyss et al., Nicollier et al., 2021] ? Should it be recalled to trace the continuity of these works?

**Equation (5)** – I agree with the definition of your **centroid frequency**. However, be careful, the definition of Thorne  is called "central frequency" and is not the same. The definition of the central frequency by Thorne is given by this equation:

$$\int_{f_1}^{f_c} P(f)df = \int_{f_c}^{f_2} P(f)df.$$

in Thorne [1986], Laboratory and marine measurements on the acoustic detection of sediment transport. I tried both formulations on several bedload data (acoustic and seismic signals) and observed that it gives different results. Maybe be add a small note on this remark ? "Note that the formulation of the central frequency [Thorne, 1986] is different from the formulation of the centroïd frequency (equation 5)."

**L249** – The term water flow velocity is not clear. Do you mean depth-averaged velocity? Velocity close to the bed? Please precise.

**L252** – Replace $V_P^{Cal}$ by $V_P^{Est}$

[Figure]

%------------------    **Results**  -------------------%

**L272-275** – "Obviously, the number of effective impacts (=real impacts + apparent impacts) for all transport modes is larger than that of the real impact" Yes, it is obvious…I don't see the point of this sentence.

**L280-283** – Can you explain this result? Is it coming from the filtering method or from the video counting?

**L301** – "Both the inclined chute experiments and the FEM simulations indicate that the impulse-mass coefficient $k_{IPM}$ varies only moderately with impact angle for a given particle size" I think you should moderate this statement as you have few data concerning the chute experiments (only two different values and comparable angles). I think more adapted to delete "Both the inclined chute experiments" in order to read "The FEM simulations indicate that the impulse-mass coefficient $k_{IPM}$ varies only moderately with impact angle for a given particle size"

**L313/315** (fig. 9b) – As previously, the experimental data are not very usable (few data and no clear trend). Should be better to read something like "The FEM simulations show that the maximum amplitude of a packet $Amp_{Max,Pac}$ increases with increasing particle impact angle θ up to about θ = 60° (Fig. 9b). The inclined chute experiments do not show a clear trend."

[Figure]

%------------------    **Discussion**  -------------------%

**L333-338** – I globally have a problem with this paragraph as I'm not convinced by the given explanations. First, for smaller particles, $r_{i,j} < 1$, it means that $N_{video} < N_{filtering\ method}$ (equation 6). You write that "This is due to the fact that in the experiment, only the particle impacts that are on the SPG plates are selected. The signal that is produced by the impacts on the concrete is dampened during wave propagation and filtered using the numerical method" Do you mean that $N_{video}$ is only computed considering the impact on the plates and that $N_{filtering\ method}$ includes some impacts that were generated on the concrete. Finally, do you mean that the filtering method is not totally efficient for the smaller diameters?

Secondly, for larger particles, $r_{i,j} > 1$ means $N_{video} > N_{filtering\ method}$ (eq. 6). You write that this "is possibly (due) because of the high impact energy generated by the large particles". I really don't see the relation between this sentence and the fact that you found less packets with the SPG method than with the video. On the contrary, I would expect that higher energy would generate a larger number of packets in the SPG signals (and so an overestimation of the number of impacts using SPG systems). Could you precise your idea?

Finally, I wonder if the definition of the equation 6 is exact, as I would expect the inverse result (underestimation of impacts with the SPG system for small diameters and overestimation for larger diameters).

**L347**: should we read "over the plate" instead of "over the channel bed" ?

**L388-390**: "A considerable difference of $Amp_{Max,Pac}$ between the transport modes could potentially be helpful in identifying sliding particles and therefore may improve the signal conversion into fractional bedload transport rates" Yes but there is also a strong dependency of $Amp_{Max,Pac}$ with bedload particle size. This dependency (on diameters and on transport modes) make the use of $Amp_{Max,Pac}$ difficult.

**L400-441**: Finally, can we conclude that the centroid frequency is a good proxy for size identification? [few dependency on transport mode and on particle velocity I guess].

---

## Author Comment (AC1)

Dear Reviewer #1,

We appreciate your valuable comments on this manuscript, all of which are important and help to improve the manuscript. We agree with your comments and have made changes in our manuscript to reflect the suggestions. Following are point-by-point responses to the comments and concerns, which we hope meet with approval. In our responses, the line number citations refer to the revised version of the manuscript.

**Comments from Reviewer #1**
- **Comment 1:** The paper describes large scale experiments with the Swiss plate geophone. Different transport modes and impact angles of bedload particles were analysed based on video recordings and geophone data.

  The paper is well written and encompasses a sound overview of research on indirect measurement methods. The experiments conducted are precise and the data analysis sound. The number of experiments is however somewhat limited as only one flow condition was analysed.

  The aim of the paper is to investigate how the signal response of the Swiss plate impacted by particles changes with transport mode. Although this aim is adequate, the application of these findings are somewhat missing. The conclusions drawn miss how a continuous transport mode measurement might help to improve bed load data collection. Is future analysis of SPG data improved and does it help to better quantify the grain size distribution?

  Why were saltation length and height or impact velocity not analysed? These are important parameters according to Sklar and Dietrich (2004) in order to estimate the impact velocity. Direct comparison to the recorded impact could have been made. Is it possible to analyse these from the video recordings?

  The number of angles used in the impact experiments is very limited with 45° and 60° only (Table 2). The FEM comparison is not convincing as it does not give similar results. A wider range of experiments would have been useful, also given the fact that 60° impact angle is very high and most of the impacts are likely much lower (e.g. Auel et al. 2017b for supercritical flow).

  Please find below line by line comments to further improve the manuscript.

  **Response:** We agree with your comments and some revisions have been made and highlighted in the revised manuscript. Our responses are as follows.

  **(1) Regarding to the application or significance of the findings of this study**

  ① Signal responses of the SPG system generated by bedload impacts can be influenced by a variety of factors, and one of the most crucial may be the complicated motion patterns during the movement of bedload particles. Therefore, it is vital to investigate the SPG response characteristics in various motion modes under the same experimental settings, with a goal to obtain a better understanding of the SPG system. For a given particle size experiment or single-size experiment, the finding of the present study can help to assess the effect of different transport modes on the variability of the SPG signal. However, the results also indicate that the transport mode cannot be precisely identified from the geophone signals in natural field conditions, due to complex interaction of different influencing factors.

  ② The investigation of this paper shows some new perspectives of the transport characteristics of the bedload particles at the Erlenbach field site, by comparing the analysis data of

transport mode from Obernach flume experiments. It indicates that for the transport conditions in the Erlenbach, saltation appears to the dominant mode for $D$ larger than about 90 mm, while for $D$ smaller than about 90 mm, the larger flow velocity at the Erlenbach could be the reason for less signal response there as compared to the Obernach flume data.

③ The centroid frequency has been found to be less sensitive to varying flow velocities than the maximum packet amplitude, based on controlled flume experiments (Wyss et al. 2016, WRR). In this study, we found that the dependences of centroid frequency on transport mode and impact angle are less than that of maximum amplitude on transport mode and impact angle. Therefore, the centroid frequency appears to be somewhat better suited for particle size identification.

④ The finding may also provide some suggestions for the design of the new filtering method, given the observation (as seen in Figure 10a) that the bedload transport mode has a lower effect on the centroid frequency $f_{\text{centroid}}$ of the SPG signals than the packet amplitude. The new filtering method developed by Nicollier et al. (2021) considers that one important information contained in $f_{\text{centroid}}$ is the particle impact location because high-frequency signals are more rapidly attenuated along the travel path than those with low frequencies. Therefore, generally, packets generated by bedload impacts outside of the Swiss geophone plates have lower $f_{\text{centroid}}$ than those triggered by impacts on a given plate. The fact that $f_{\text{centroid}}$ shows less dependency on transport mode supports our confidence in the filtering method.

**(2)  Why were saltation length and height or impact velocity not analysed?**

Auel et al. (2017) defined the probability for the rolling mode as the ratio of the travelled distance by a rolling particle to the overall distance determined by the sum of saltation and rolling modes averaged over the numbers of particles travels. Comparably, in our study, the probability is calculated as the number of signal packets generated by particle impacts for each transport mode divided by the total number of effective packets. The two definitions are somewhat different but our experimental observations did not allow to use the same definition as Auel et al. (2017). With regard to impact velocity, the limitation of the video recording and that the set-up was not designed for evaluating its value.

**(3)  Limited number of slope angles used in the impact experiments**

Significant differences between transport modes (saltation, rolling, and sliding) were observed with regard to the impact angle on the channel bed. Therefore, an inclined chute experiment was conducted in still water to examine the effect of particle impact angle on the signal response of the SPG system (Fig. 1c). Due to the friction, it was difficult for the particles released at the top of the wood chute to keep moving at small chute angles. Hence, the experimental angles in this study were chosen to be 45° and 60° for bedload particles with variable sizes, and no smaller impact angles could be investigated. Therefore, the investigated angles for the chute experiments were rather steep compared to real particle impact angles. Hence FEM simulations were used to analyze the effect of different bedload impact angles, covering a full range of angles (from 0° to 90°). The advantage of the FEM method is that there are not many interfering factors and thus the variables used in the model can be well controlled.

- **Comment 2:** Line 80, 81. Please don't use three headings without text.

  **Response:** Agreed. Some content has been added between the headings 2 and 2.1:

  "In the methods section, we introduce in turn the controlled experiments including controlled flume experiments and inclined chute experiments, numerical simulations with the FEM model, methods of transport mode analysis and signal processing." (L81-82)

- **Comment 3:** Line 87: Are MPA and JPM data used for further analysis in this paper?

  **Response:** The MPA data has been used to measure the transport velocity of bedload, combined with the SPG system, as explained in L285-294 and Appendix B (L575-612). Data recorded by the JPM system was not used in this study.

- **Comment 4:** Line 92: bedload sampled in the field. Where is this bedload from? Is there any relation to a real river. This should be mentioned here. Why $D_{67}$ and $D_{84}$? Please state their values here.

  **Response:** Thanks for this valuable advice. The channel bed of the flume facility was reconstructed considering the bed characteristics corresponding to the Navisence field site in Switzerland. The bed roughness is made up by gravel particles that have a size corresponding to $D_{67}$ and $D_{84}$ of the bedload material sampled in the Navisence. The characteristic grain sizes $D_{67}$ and $D_{84}$ were chosen in order to allow all experimental particles to pass smoothly through the reconstructed channel bed with a similar roughness as the natural bed. In addition, the flow velocity and flow depth were adjusted to match with that in the field site. Information on bed characteristics and hydraulic conditions are given in **Table 1.**

  **Table 1:** Bed and flow conditions at the Navisence field site and in the flume experiments.

  | Parameters | Units | Value |
  |---|---|---|
  | Bed surface $D_{67}$ | mm | 180 |
  | Bed surface $D_{84}$ | mm | 280 |
  | Flow depth (Navisence field site) | m | 0.4-0.65 |
  | Flow depth over the SPG (flume) | m | 0.54 |
  | Flow discharge (flume) | $m^3\ s^{-1}$ | 1.78 |
  | Flow discharge (Navisence field site) | $m^3\ s^{-1}$ | 1.2-2.28 |
  | Flow velocity (Navisence field site) | $m\ s^{-1}$ | 3-3.5 |
  | Flow velocity 0.1 m above the SPG plates (flume) | $m\ s^{-1}$ | 3.30 |
  | Flume gradient of the natural bed | % | 4 |
  | Flume width | m | 1.02 |
  | Froude number (flume) | - | 1.43 |
  | Froude number (Navisence field site) | - | 1.39-1.51 |

  Some related statement and the table shown above have been added in the revised manuscript (L110-111, L113-114, L125).

- **Comment 5:** Line 107. Please indicate the discharge and the Froude number, the flow is supercritical.

  **Response:** Please see the response to the comment 4 (above).

- **Comment 6:** Line 111: The video images in Fig 1b are distorted. Did you apply any correction?

  **Response:** The distortion of the video images was caused by the wide-view mode used during recording. However, in the present study, we only focus on the impact instants of particles on the bed. For

the dimensions during particles movement, we didn't apply a correction on the video images.

- **Comment 7:** Line 140: speed. The discussion is a little academic if the word *speed or velocity* should be used as speed is refers to a scalar and velocity to a vector. As you use the term velocity mostly throughout your paper (e.g. caption Fig 2 on the same topic), you should use it consistently.

  **Response:** We accept this suggestion and have changed the word "speed" to "velocity" accordingly throughout the manuscript (L152, L321).

- **Comment 8:** Line 125: The chosen angles are rather steep compared to real particle impact angles as e.g. given in Auel et al. 2017b (not cited). Note that these angles are flat due to supercritical flow conditions. Your impact angles might be larger. Please elaborate if 45 and 60° are close to your observations.

  **Response:** Thanks for this suggestion. The purpose of the inclined chute experiments can refer to the previous responses above (response to comment 1). The investigated angles in this study were 45° and 60° for natural bedload particles with sizes ranging from 12.3 mm to 95.5 mm and for spherical particles with sizes ranging from 20 mm to 82 mm. It's true that the chosen angles are rather steep compared to real particle impact angles, and angles of 45° and 60° cannot fully represent actual observed values yet. Hence the FEM simulation is used for non-vertical impacts to investigate the effect of different bedload impact angles, covering a full range of angles (from 0° to 90°) that can be observed in the flume experiments.

  Some statements have been added in the text (L137-141).

- **Comment 9:** Line 169: studies have shown.

  Line 169: the probability of transport mode. The transport mode has no probability, saltation, rolling or suspension have a probability. Please rewrite.

  **Response:** Agreed. This sentence has been rephrased in the manuscript (L194).

- **Comment 10:** Line 171: The choice of Teta_crit is quite crucial as you use it for plotting your results as a function of transport stage T and compare it with many other results taken from Auel et al 2017. Please elaborate in detail how you derived this value. You have a fixed bed. What did Schneider and Shahmohammadi use in their studies? Auel et al. 2017 list a large variety of these values in their Table 3 and state: *most laboratory studies investigated motion of isolated particles moving on top of a bed of similar roughness size, for which θc = 0.008 to 0.01 were determined (Fenton and Abbott, 1977; Dancey et al.,2002).*

  You also investigated isolated particles over a fixed bed of similar roughness size. Therefore, the choice of 0.03 is too high from my point of view. At least you should perform a sensitivity analysis of your results by varying teta_crit in order to check the effects on your results.

  **Response:** Thanks to the reviewer for these careful comments. Indeed, it is quite crucial and difficult to obtain the critical Shields parameter $\Theta_{Critical}$ precisely. Auel et al. (2017) list some values of $\Theta_{Critical}$ based on a large number of lab studies, indicating that $\Theta_{Critical}$ = 0.008 to 0.01 were determined (Fenton and Abbott, 1977; Dancey et al.,2002) in the case of motion of isolated particles moving on top of a bed of similar roughness size. However, $\Theta_{Critical}$ in the present study should have a larger value than 0.008 to 0.01 due to different bed roughness conditions.

  **(1) Bed roughness**

For a flat bed, the bed roughness height is given in terms of the Nikuradse roughness height $k_s$ (Camenen et al. 2009). As stated in Camenen et al. (2009), $k_s$ is expected to be in the order of magnitude of the median grain diameter or of some larger grain size percentiles ($k_s$ = 1 to 5 times $D_{50}$, $D_{65}$, $D_{84}$, or $D_{90}$ according to the literature) of the bed. In Auel et al. 2017, the equivalent sand roughness height corresponding to $k_s$ has been used, which is determined as $k_s$ = 0.20 mm. In present study, $k_s$ = 180 mm to 280 mm because the bed surface $D_{67}$ and $D_{84}$ equal to 180 mm to 280 mm, respectively.

Subsequently, an effective roughness ratio $k_s/D$ is given by a value of the effective roughness height $k_s$ divided a characteristic grain size D. In Auel et al. 2017, the effective roughness ratio $k_s/D$ ranges from 0.038 to 0.011. Then the effective roughness ratio $k_s/D$ in our study is determined ranging from 14.63 to 1.05, if $k_s$ = 180 mm is taken, which is higher than that in Auel et al. 2017. Studies have shown that $k_s/D$ and the total Shields number $\theta$ show a positive correlation (Camenen et al. 2009), based on a large number of experiments and theories. Consequently, the higher value of $k_s/D$ makes the Shields number $\theta$ of our study greater than that in Auel et al. 2017. Thus, the critical Shields parameter $\Theta_{Critical}$ should also be higher than that given in Auel et al. 2017.

**(2)  $\Theta_{Critical}$ determined by the largest particles**

It can be assumed that the maximum particle size used in our experiments (which was transported for the given flow conditions) was close to (but not equal to) the critical size of bedload particles that start moving during the experiments. Therefore, it is possible to estimate $\Theta_{Critical}$ based on the maximum particle size used in our experiments.

$$\Theta_{Critical} < \frac{\overline{\tau_b}}{(\rho_s - \rho)gD_{Max}} = 0.0365$$

where $\overline{\tau_b}$ is the (time-averaged) bed shear stress. According to this calculation, for our experimental conditions $\Theta_{Critical}$ should be smaller than 0.0365.

**(3)  Comparation with other field and flume studies**

Considering that the controlled experiments in this study were performed in a flume facility replicating the field site, the critical Shields parameter $\Theta_{Critical}$ should be similar to that of the field site. According to a study for serval mountain streams (Schneider et al. 2015), the median value of the effective shear stress (corresponds to $\Theta_{Critical}$) is determined to be about 0.03 from the main dataset, showing less dependency with the slope of the stream bed.

[Figure]

**Figure 3.** Dimensionless reference shear stress ($\tau^*_{rD50}$) related to channel bed slope for (a) the *total* acting bed shear stress $\tau^*_{rD50}$ and (b) the reduced *effective* shear stress $\tau^{*'}_{rD50}$ (RR2011). Filled circles correspond to $\tau^*_{rD50}$ values from analyzing fractional transport rates (see reference approach section 23.2 and Figure 2b. Crosses indicate $\tau^*_{rD50}$ derived from analysis of total dimensionless transport rates. The thick black line was fitted to the Main Data set, and the thick gray line to the HS Data set, for $\tau^*_{rD50}$ derived from the analysis fractional transport rates. In Figure 3a, the empirical relations of *Bunte et al.* [2013], *Lamb et al.* [2008], and *Mueller et al.* [2005] are given. The dashed blue line in Figure 3b corresponds to the median $\tau^{*'}_{rD50}$ from Main Data set.

Figure in Schneider et al. (2015, WRR)

Shahmohammadi et al. (2021) statistically obtained $\Theta_{Critical}$ - Relative Roughness correlation curves from the data of a large number of flume experiments. The Relative Roughness of our experiments ranges from 0.023-0.32, resulting in a median value for the critical Shields parameter of approximately 0.05.

Given the fact that the experimental conditions in this paper are more comparable to those described by Schneider et al. (2015), the critical Shields parameter $\Theta_{Critical}$ in our flume experiments is assumed to be 0.03.

Some statements have been added in the manuscript (L180-191).

- **Comment 11:** Line 157: you mention uniform flow conditions here. This should be mentioned in chapter 2.1.2 already, if the flow is uniform or gradually varied.
  **Response:** This has been added in section 2.1.2 (L111).

- **Comment 12:** Line 176 ff. Chapter 2.3.2. It is not clear for what the description of the forces are used. Fig 4b and c are not explained at all. These forces have to be elaborated in more detail here and used also in the discussion section. See Auel et al 2017b for discussion on vertical and horizontal energy transfer. Else Fig 4 and the corresponding text should be deleted.
  **Response:** Thanks for this comment. The reason for showing these forces here is simply to better illustrate the determination of the impact instant (at $T_1$) during the video analysis. These force couples act together on the particle, and finally rotate the particle. This small rotation of the bedload particle occurs immediately after impacting, allowing to determine the impact instant (at $T_1$) from the video frames. Therefore, the forces that are described in Fig. 4 were not measured and cannot be measured under the present experiment set-up.

- **Comment 13:** Line 184: please elaborate more what the vertical support force is? That is not entirely clear for me. Is this connected to the lift force? The lift force is caused by both the flow velocity gradient (Saffman force) and the spinning motion of the particle (Magnus force).
  Please elaborate more on that.
  **Response:** In order to better describe how we obtain the time instant when a particle impacts onto the channel bed and the plates during the video analysis, we present three sketches of transport modes of saltation, rolling and sliding, respectively, and also indicate an interaction between the bedload particle and the SPG plate. The forces in these sketches are used only as an aid to illustrate how we observe a few moments when the particles are in contact with the plate or the channel bed. Therefore, the support force here refers to the vertical component force of the plate acting on the contacted particle at this moment. Consequently, we take the instant when the particle undergoes a small rotation as the moment of impact. Hence, we keep the Figure 4 in the manuscript.

[Figure]

We have added some contents in the manuscript (L204-205).

- **Comment 14:** Line 252: I guess you mean V$_P$$^{EST}$ here instead of V$_P$$^{CAL}$. Julien and Bounvilay analysed rolling particle velocities. You have mostly saltation in your experiments. Auel et al. 2017 found that particle velocity is only 8.5% lower than flow velocity for saltation in supercritical flows (hence r$_{pw}$ = 0.915). Finally which value for r$_{pw}$ did you chose for further analysis as 0.3 to 0.8 is a large variation.

  **Response:** Thanks for this valuable comment on the particle velocities. Yes, it should be $V_P^{Est}$ (i.e. the estimated particle velocity in present study). This has been revised in the manuscript (L286).

  To compare with other studies, the estimated particle velocities with the $r_{PW}$ ranging from 0.3 to 0.8 are shown in Fig. 12b (the red area). It is true that $V_P^{Est}$ is slower than that given by Auel et al. 2017. However, Auel's estimate that the saltation particles velocity is only 8.5% lower than flow velocity in supercritical flows, is based on his flume experiments, for which the effective roughness ratio k$_s$/D ranged from 0.038 to 0.011. The effective roughness ratio k$_s$/D in our experiments ranged from 1.05 to 14.63, and thus there was likely a larger relative difference between particle and flow velocity in our case than in Auel's experiments.

  To obtain the bedload velocity more precisely, we calculated the particle velocities $V_P^{Cal}$ using the arrival time difference between the two monitoring systems (SPG and MPA) and corresponding particle travel distances (illustrated by the blue triangles in Fig. 12b). The calculated results showed that the ratio between particle velocity and flow velocity ranges from 0.53 to 0.88.

  See line 517 in the revised manuscript.

- **Comment 15:** Line 285: Fig 7. No need to do a semi log plot here, better use a regular Y axis.

  **Response:** Agreed. Figure 7 has been revised based on the reviewer's comment (L315).

- **Comment 16:** Line 293, line 308, line 318, Fig 8b, Fig 9b, Fig. 10b. Please elaborate more on the difference between FEM and lab experiments. For me it looks like the results do not match at all. With the FEM I would expect the you reach results close to the still water experiments. How did you calibrate the FEM model?

  **Response:** Significant differences between transport modes (saltation, rolling, and sliding) were observed with regard to the impact angle on the channel bed. Therefore, an inclined chute experiment was conducted in still water to examine the effect of particle impact angle on the signal response of the SPG system (Fig. 1c). However, the investigated angles are rather steep compared to more realistic particle impact angles to be expected in field conditions and in our flume experiments, and thus the angles of 45° and 60° may not be very representative. Hence the FEM simulation was used as main method to investigate the effect of different bedload impact angles, covering a full range of angles (from 0° to 90°), on the SPG signal response. The FEM results were compared with the observations from the inclined chute experiment for the cases of 45° and 60°. To give less weight to the chute experiment data covering a very limited range of slope angles, they were removed from Fig 8b, Fig 9b and Fig. 10b, and are now shown in Table 5 (L324-327). While there are discrepancies between the chute experiment data and the FEM results for the values shown in the (old) Fig 8b, Fig 9b and Fig. 10b, the limited change of the characteristic values of the chute experiments with changing slope angle are in qualitative agreement with the FEM

results with approximately constant characteristic values over a much larger range of slope angles from 20° to 90°. The discrepancy especially that the values of $Amp_{Max,Pac}$ for the FEM simulations are considerably larger than those from the chute experiments for the impact angles of 45° and 60°. This may be partly because that the impact velocities in the inclined chute experiments were overestimated (L363-365, L462-464).

Before performing the numerical simulations, the FEM model had been calibrated with results obtained from the previous lab experiments (drop tests) with quartz spheres (see below, Chen et al. 2021), which is not shown in the present paper. To make this part more understandable, some details about FEM model have been added in Appendix A (L150-151, L558-574).

[Figure]

**Figure A1:** (a) Side-view photo (Gaillard, 2018) and (b) sketch (taken from Chen et al. 2021) of the drop-test set-up used in earlier laboratory tests to measure the impact velocity. $h$ is the distance between the bottom surface of the sphere and the plate, and $d$ is the distance between the bottom of the Plexiglas tube and the plate. The tube protects the sphere from flow turbulences in cases where the set-up is used at field sites. (c) Comparison of the FEM signal and the experimentally generated signal (from the drop test) in the Z-direction (perpendicular to the plate and pointing up), triggered by a single bedload particle (diameter $D = 120$ mm) impacting on the centric location of the plate with a velocity of 0.777 m/s.

- **Comment 17:** Line 329 ff; Discussion 4.1 should be improved to better understand how these results help to improve the geophone data analysis.

  **Response:** Thanks for this comment. This part of discussion has been improved.

● **Comment 18:** Line 330: Please explain what $r_{i,j}$ is good for? For what do you use or need this parameter?

**Response:** $r_{i,j}^{Packet,V\_F}$ is actually the ratio of the packet counts by two different methods. Specifically, $r_{i,j}^{Packet,V\_F}$ is the ratio of the total number of real packets over all transport modes based on the video observations to the real-packet number determined by the filtering method, which can be calculated by

$$r_{i,j}^{Packet,V\_F} = \frac{N_{i,j}^{Packet,V}}{N_{i,j}^{Packet,F}} \quad \text{(Eq.6 in the manuscript)}$$

where $N_{i,j}^{Packet,V}$ is the total number of real packets for experimental run $i$ and grain-size class $j$ over transport modes based on the video analysis; $N_{i,j}^{Packet,F}$ is the number of real packets for experimental run $i$ and grain-size class $j$, determined by the filtering method.

The purpose is simply to cross-check the results of packet counts and make the data more plausible.

Some related contents are given in the manuscript (L265-270, L374-383).

● **Comment 19:** Line 396. Explain the Hertz theory in a few words please.

**Response:** Agreed. According to the Hertz contact theory (Johnson, 1985; Thorne, 1986), the frequency at which the geophone plate vibrates is controlled by the contacting particle size (Bogen & Møen, 2003; Barrière et al., 2015; Rickenmann, 2017), indicating that the characteristic frequency decreases with increasing contacting particle size.

Some explanations have been added in the text (L442-445).

● **Comment 20:** Line 428: It should be noted here, that Auel et al 2017 did not differ between sliding and rolling. Both modes are included in their rolling mode.

**Response:** Thanks for this comment. This point has been added in the manuscript (L484-485).

● **Comment 21:** Line 436: Difference of estimation of rolling probability of Auel and you is not clear. Please rewrite, how do you obtain your value and what the difference to Auel is.

**Response:** According to Auel et al., 2017, the definition of the probability for the rolling mode is the ratio of the travelled distance by a rolling particle to the overall distance determined by the sum of saltation and rolling modes averaged over the number of particles travels.

Comparably, in our study, the probability for each transport mode is considered as the fraction/ratio that is calculated by the number of signal packets (generated by particle impacts) for each mode to the total number of packets.

This corresponding sentence has been rephrased in the manuscript (L491-494).

● **Comment 22:** Line 445. … Auel et al. (2017) indicated that large particles have a high probability $P_{Rol}$. It is important to note that this is true for similar transport stages T (as T is dependent on friction velocity and particle size).

**Response:** This has been noted in the text (L502).

● **Comment 23:** Line 446: Unclear. Please rephrase. Energy consumption of small particles is larger, that is why they saltate more? By energy consumption you mean energy transfer to the particle?

**Response:** To avoid misunderstanding, this sentence has been deleted in the manuscript.

● **Comment 24:** 447: Proll decreases with large sizes? Please refer to the respective Figure in your results (12a?, In 12a, almost no variation of PR is visible for your saltation results). If Proll decreases, consequently Psal increases. Why should this be the effect of gravity? This result remains unclear. Please define how the 3 modes are related in your analysis (e.g. PRoll = (1-Psal), etc.)

**Response:** Thanks for this comment. We included our experimental data in Fig. 12a (Line 525) by defining the cumulative probabilities $P_{Sli} + P_{Rol} + P_{Sal} = 1$. For the revision, we have first corrected a wrong plotting of our data in Fig. 12. We changed the discussion text as follows: "For the three smallest $T$ values our data show that the sum $(P_{Rol} + P_{Sli})$ values are somewhat smaller whereas $P_{Sli}$ is slightly larger than for other (higher) $T$ values. For small $T$ values the bed shear stress is very close to that of incipient motion of particles, and for more angular or flatter-shaped particles this might have caused a decrease in the $P_{Rol}$ values. Indeed, flatter-shaped particles are more likely to move in the sliding mode according to our video observations. For the four largest $T$ values, the rolling and saltation particles of our experimental data are reasonably consistent with the data of Auel et al. (2017a)."

Some statements have been added in the manuscript (L502-507).

● **Comment 25:** 448f: As transport stage T is a non-dimensional parameter, it should not play a role if your particles are larger.

**Response:** Agreed. We have modified the discussion regarding our experimental data in Fig. 12a (L500-507).

● **Comment 26:** 451: note that Auel et al 2017 did not distinguish between rolling and sliding (see comment line 428).

**Response:** Thanks for this comment. This point has been added in the manuscript (L484-485).

● **Comment 27:** 454: The proposed line between sliding and rolling is interesting. Is this really a fit? Please state R^2.

**Response:** The previous fitted model was obtained by linear regression, with $R^2 = 0.5$. We have removed the line plotted in the Fig 12a (L525).

● **Comment 28:** 455: I agree, that more flow conditions would be needed. Your variation in transport stage stems from different particle sizes but not flow (friction) velocities.

The meaning of $V_P^{Est}$ is not clear for me. A variation between 30 and 80% is very large. What do you want to show with that?

**Response:** Generally, the value of bedload particle velocity $V_P$ is expected to be less than the water flow velocity $V_W$. The ratio $r_{PW} = V_P/V_W$ is given through a larger number of experiments and variable experimental conditions, ranging from 0.3 to 0.8 for natural particles as suggested by Julien and Bounvilay (2013). Hence the bedload transport velocity can be estimated empirically from the water flow velocity. In this study, we calculate $V_P^{Est}$ only for comparison with other experimental data and our data.

● **Comment 29:** 467: Given that this is a double log plot, the data is not close to Auel et al. 2017. Your power

<table>
<tr><td></td><td>function is 0.32 while it is 0.5 for Auel.</td></tr>
<tr><td>**Response:**</td><td>Thanks for this comment. It is true that the power function obtained by our flume experiments data is 0.32 while it is 0.5 in Auel et al. (2017). We observe that this variability mainly results due to the smallest value of $T$, demonstrating that the calculated velocity values deviate from the Auel's model (Auel et al. 2017). The reason for this might be that when the particles are getting larger, $T$ does not play a major role, as previously stated.</td></tr>
</table>

To make it clear, we removed the model line of our study in Fig.12b. Some relevant statements have been excluded in the manuscript (L525, Fig. 12b).

- **Comment 30:** 482 Conclusion.

  Please indicate what would be the benefit of these statements for the geophone data analysis. Does your analysis help to indicate the transport mode with the SPG in the field? This would be the main result of your study. However, your discussion and conclusions do not really reveal if this is possible with your results.

  **Response:** Thanks for this comment. Some statements have been added in the text. Also see the responses of the comment 1.

**Additional clarifications**

In addition to the above comments, spelling and grammatical errors pointed out by the reviewer have been corrected in the manuscript.

We look forward to hearing from you in due time regarding our submission and to respond to any further questions and comments you may have.

Sincerely,
The authors of manuscript esurf-2021-72.

---

## Author Comment (AC2)

Dear Dr. Thomas Geay,

We appreciate your valuable comments on this manuscript. All the comments are important and help to improve this manuscript. We agree with your comments and have been able to incorporate changes to reflect the suggestions. We have highlighted the changes within our manuscript.

Here is a point-by-point response to the comments and concerns, which we hope meets with your approval. In our responses, the line number citations refer to the revised version of the manuscript.

**Comments from Reviewer #2**

- **Comment 1:** This paper deals with the use Swiss Plate Geophone (SPG) for bedload monitoring and it studies the effect of transport modes (rolling, sliding, saltating particles) on SPG signals. This work gives a new understanding on how bedload particles are transported and how it affects the signals monitored by surrogate methods. In the introduction, the scientific question is well presented, with a nice overview of the existing literature. I also appreciated to read the method section as the description of the setups and notations are very clear. The results and discussion sections are also well written. We can learn how several parameters of SPG signals are affected by transport modes, bedload diameters, angle of impact, ect. These results help to understand the variability of the SPG calibration curves obtained in field experiments. These results can be extended to study the behaviour of other bedload monitoing technics that are in development (seismic, acoustic methods for example).

  Finally, I recommend this paper with minor revisions and thank the authors for their careful job. Some suggestions and remarks are listed in the supplement file, joined to this comment.

  **Response:** Much appreciated of your valuable comments which are very helpful to improve our manuscript. We agree with your comments, and some revisions have been made and highlighted in the revised manuscript. The responses to the comments are presented as follows.

- **Comment 2:** **Introduction. L39.** Could be valuable to add other papers using direct calibration. For example: Bakker, M., Gimbert, F., Geay, T., Misset, C., Zanker, S., & Recking, A. (2020). Field Application and Validation of a Seismic Bedload Transport Model. Journal of Geophysical Research: Earth Surface, e2019JF005416. https://doi.org/10.1029/2019JF005416

  Geay, T., Zanker, S., Misset, C., & Recking, A. (2020). Passive Acoustic Measurement of Bedload Transport: Toward a Global Calibration Curve? Journal of Geophysical Research: Earth Surface, 125(8), 1–19. https://doi.org/10.1029/2019JF005242

  Rennie, C. D., Vericat, D., Williams, R. D., Brasington, J., & Hicks, M. (2017). Calibration of acoustic doppler current profiler apparent bedload velocity to bedload transport rate. Gravel-Bed Rivers: Process and Disasters, 209–233. https://doi.org/10.1002/9781118971437.ch8

  **Response:** Thanks for the valuable advices. We agree that it is worthwhile to cite more papers using direct calibration in the manuscript (L33-35, L646-648, L702-703, L752-754).

- **Comment 3:** **L56-58.** To my mind, Thorne [1985] has not shown that transport modes are associated to

different acoustic response (it is only suggested in this paper). Please check my remark and remove this reference if this review is true.

**Response:** Agreed. This citation has been removed in the manuscript.

● **Comment 4:** **Methods. Table 1** - When monitoring bedload, we are interested in estimating bedload fluxes (in kg/min, g/m/s or else). In the presentation of the Flume experiments (section 1.1.2), I think important to have an estimation of these bedload fluxes (even if imprecise). I believe that the number of moving particles strongly affects the modes of transport and their associated characteristics (impact velocities/angles, etc.), due to interaction between particles. This was suggested in the following paper: Gimbert, F., Fuller, B. M., Lamb, M. P., Tsai, V. C., & Johnson, J. P. L. (2018). Particle transport mechanics and induced seismic noise in steep flume experiments with accelerometer-embedded tracers. Earth Surface Processes and Landforms. https://doi.org/10.1002/esp.4495. The number of moving particles is dependent on bedload fluxes and bedload sizes. For me, it is therefore very important to give an overview of the bedload fluxes used in these experiments.

**Response:** Thanks for this valuable advice. We agree that the number of moving particles can strongly affect the modes of transport, and the number of moving particles is dependent on bedload fluxes and bedload sizes. In the present study, we focus on each particle impact on the bed or on the geophone plates.

The bedload particles were released into the flume several meters upstream of the SPG system with known particle numbers and masses for each experimental run (see Section 2.1.2). The transport velocity of bedload particles for all grain size classes was calculated ranging from 1.74 m/s to 2.91 m/s with a mean value of 2.08 m/s, resulting in the bedload fluxes estimated in our study ranging from 0.092 kg/m/s to 20.31 kg/m/s. We have added and expanded Tables (Tabs. 1 and 2) with information on both flow conditions in the flume and on unit bedload transport rates.

Some related statements have been added in the revised manuscript (L120-121, L126).

● **Comment 5:** **L128-130** - What are the average impact velocities of the particles in the flow (i.e. in the flume experiments)? Is it comparable to the particle velocity in the inclined chute experiments? Please add a sentence on this comparison.

**Response:** The particle impact velocity in the inclined chute experiments was estimated to be much higher than the average impact velocities (fractions of a meter per second) of the particles in the flume experiments. This is because the potential height of the particles released on the chute bed is considerably higher than the particle hop height in the flume experiments. The inclined chute experiments were performed here only to investigate the effect of (two) impact angle(s) on the SPG signal response.

Some related statements have been added in the revised manuscript (L137-141).

● **Comment 6:** **Figure 2b.** Add "view" to read "Cross-sectional view of the FEM model of the SPG system."

**Response:** Agreed. This sentence has been revised in the manuscript (L156).

● **Comment 7:** **L154** - Maybe delete "the" from "the bedload particles".

**Response:** Agreed. This definite article has been deleted in the sentence.

● **Comment 8:** **L210** –is $k_{IPM}$ comparable to the $k_b$ of almost all other studies [Rickenmann et al., 2013; Wyss et al., Nicollier et al., 2021]? Should it be recalled to trace the continuity of these works?

**Response:** The number of impulses $I$ is found to be reasonably well correlated with the total transported bedload mass $M_{Tot}$ (see below) based on direct bedload measurements at various field sites.

$$I = k_b M_{Tot}$$

where $k_b$ is the site-dependent calibration coefficient. The coefficient $k_b$ is further developed for different grain size classes $j$ as the coefficient $k_{bj}$, which has been utilized to infer bedload transport by grain-size fraction from the SPG signal (Wyss et al., 2016c; Nicollier et al., 2020).

The mass-impulse coefficient $k_{IPM}$ given in the present study is similar to the coefficient $k_b$ in other studies (Rickenmann et al., 2013; Wyss et al., Nicollier et al., 2021) but has not exactly the same definition. $k_{IPM}$ is more comparable to the $k_{bj}$ value, although not completely the same. The $k_{bj}$ values also include the mass of non-impacting particles, while $k_{IPM}$ is defined as the number of impulses (triggered by each impact) per particle mass in present study. Note that the particle impact locations (concrete, plate G1, plate G2, or plate boundaries) and impact instants are determined during our video analysis, and the signal impulses used for the calculation of $k_{IPM}$ are extracted from the real packets. Therefore, the propagating signal noises caused by apparent impacts have been excluded in our analysis, resulting in the different definitions between $k_{IPM}$ and $k_{bj}$.

$$k_{IPM} = \frac{I}{M}$$

where $I$ is the number of signal impulses recorded by the SPG system and $M$ is the corresponding mass of the transported particle.

Some related statements have been added in the revised manuscript (L231-236, L319-320).

● **Comment 9:** **Equation (5)** – I agree with the definition of your **centroid frequency**. However, be careful, the definition of Thorne is called "central frequency" and is not the same. The definition of the central frequency by Thorne is given by this equation:

$$\int_{f_1}^{f_c} P(f)df = \int_{f_c}^{f_2} P(f)df$$

in Thorne [1986], Laboratory and marine measurements on the acoustic detection of sediment transport. I tried both formulations on several bedload data (acoustic and seismic signals) and observed that it gives different results. Maybe add a small note on this remark? "Note that the formulation of the central frequency [Thorne, 1986] is different from the formulation of the centroid frequency (equation 5)."

**Response:** Thanks for this valuable comment on the calculation of (central) frequency. We agree and the sentence suggested by the reviewer has been added in the text (L246-247).

● **Comment 10:** **L249** – The term water flow velocity is not clear. Do you mean depth-averaged velocity? Velocity close to the bed? Please precise.

**Response:** In relation to Eq. 9 in this paper, the water flow velocity $V_W$ means the depth-averaged flow velocity based on Julien and Bounvilay (2013). Comparably, the flow velocity in our flume experiments was measured by using a flow meter (OTT MFpro) positioned 0.1 m above the SPG

plate in the middle of the cross-section.

The term "depth-averaged" has been added in the corresponding sentence (L280).

● **Comment 11:** **L252** – Replace $V_P{}^{Cal}$ by $V_P{}^{Est}$

**Response:** Thanks for this comment. "$V_P^{Cal}$" in this line should read "$V_P^{Est}$", which is the estimated particle velocity in the present study.

This sentence has been revised based on the reviewer's comment (L283).

● **Comment 12:** **L272-275** – "Obviously, the number of effective impacts (=real impacts + apparent impacts) for all transport modes is larger than that of the real impact" Yes, it is obvious…I don't see the point of this sentence.

**Response:** This is the first time the term "effective impacts" appears in the article, so we defined it in the text. This sentence has been rephrased in the manuscript (L303-304).

● **Comment 13:** **L280-283** – Can you explain this result? Is it coming from the filtering method or from the video counting?

**Response:** $r_{i,j}{}^{Packet,V\_F}$ is actually the ratio of the packet counts using two different methods. It is defined as the ratio of the total number of real packets over all transport modes based on the video observations to the real-packet number determined by the filtering method:

$$r_{i,j}^{Packet,V\_F} = \frac{N_{i,j}^{Packet,V}}{N_{i,j}^{Packet,F}} \quad \text{(Eq.6 in the manuscript)}$$

where $N_{i,j}{}^{Packet,V}$ is the total number of real packets for experimental run $i$ and grain-size class $j$ over transport modes based on the video analysis; $N_{i,j}{}^{Packet,F}$ is the number of real packets for experimental run $i$ and grain-size class $j$, determined by the filtering method.

The purpose is simply to cross-check the results of packet counts and make the data more reliable.

Some related contents have been added to the manuscript (L265-270, L374-384).

● **Comment 14:** **L301** – "Both the inclined chute experiments and the FEM simulations indicate that the impulse-mass coefficient k$_{IPM}$ varies only moderately with impact angle for a given particle size" I think you should moderate this statement as you have few data concerning the chute experiments (only two different values and comparable angles). I think more adapted to delete "Both the inclined chute experiments" in order to read "The FEM simulations indicate that the impulse-mass coefficient k$_{IPM}$ varies only moderately with impact angle for a given particle size"

**Response:** Agreed. This sentence has been rephrased based on the reviewer's comment (L338).

● **Comment 15:** **L313/315** (fig. 9b) – As previously, the experimental data are not very usable (few data and no clear trend). Should be better to read something like "The FEM simulations show that the maximum amplitude of a packet Amp$_{Max,Pac}$ increases with increasing particle impact angle θ up to about θ = 60° (Fig. 9b). The inclined chute experiments do not show a clear trend."

**Response:** Thanks for this comment and we agree. This sentence has been rephrased in the text (L348).

● **Comment 16:** **L333-338** – I globally have a problem with this paragraph as I'm not convinced by the given

explanations. First, for smaller particles, $r_{i,j} < 1$, it means that $N_{video} < N_{filtering}$ method (equation 6). You write that "This is due to the fact that in the experiment, only the particle impacts that are on the SPG plates are selected. The signal that is produced by the impacts on the concrete is dampened during wave propagation and filtered using the numerical method" Do you mean that $N_{video}$ is only computed considering the impact on the plates and that $N_{filtering}$ method includes some impacts that were generated on the concrete. Finally, do you mean that the filtering method is not totally efficient for the smaller diameters?

Secondly, for larger particles, $r_{i,j} > 1$ means $N_{video} > N_{filtering}$ method (eq. 6). You write that this "is possibly (due) because of the high impact energy generated by the large particles". I really don't see the relation between this sentence and the fact that you found less packets with the SPG method than with the video. On the contrary, I would expect that higher energy would generate a larger number of packets in the SPG signals (and so an overestimation of the number of impacts using SPG systems). Could you precise your idea?

Finally, I wonder if the definition of the equation 6 is exact, as I would expect the inverse result (underestimation of impacts with the SPG system for small diameters and overestimation for larger diameters).

**Response:** For small particles, $r_{i,j}^{Packet,V\_F} < 1$ means that the total number of real packets for experimental run $i$ and grain-size class $j$ over all transport modes based on the video analysis $N_{i,j}^{Packet,V}$ is smaller than that determined by the filtering method $N_{i,j}^{Packet,F}$. $N_{i,j}^{Packet,V}$ is counted only considering the impacts on the plates. Considering that the signals triggered by small particle impacts are dampened quickly during wave propagation, the filtering method in fact does not have a significant improvement for the identification of real packets caused by small particles. Therefore, one reason for $r_{i,j}^{Packet,V\_F} < 1$ could be the limitation during the video observation and analysis: Limited visibility due to flow turbulence can cause an underestimation of the number of impacts on the SPG plates, because some small particles that impact on the plate boundaries on the other side cannot be easily observed.

For large particles, $r_{i,j}^{Packet,V\_F} > 1$ means that $N_{i,j}^{Packet,V} > N_{i,j}^{Packet,F}$. There may be several reasons:

a) Some particles that impacted close to boundaries (e.g. bolts) of the geophone plates were filtered out.

b) The number of impacts caused by sliding particles increases as the particle size increases. However, some sliding particles could be incorrectly filtered out due to weak impact amplitude/energy.

c) Simply due to the filtering method itself. It was found that some bedload particles can be misclassified in the largest size classes using the filtering method.

Some related contents have been added to the manuscript (L374-377, 379-383).

- **Comment 17:** L347: should we read "over the plate" instead of "over the channel bed"?
  **Response:** Agreed. This has been revised in the text (L391).

- **Comment 18:** L388-390: "A considerable difference of $Amp_{Max,Pac}$ between the transport modes could potentially be helpful in identifying sliding particles and therefore may improve the signal conversion into fractional bedload transport rates" Yes but there is also a strong dependency of $Amp_{Max,Pac}$ with bedload particle size. This dependency (on diameters and on transport modes)

make the use of Amp$_{\text{Max,Pac}}$ difficult.

**Response:** That's a good comment. For a given size experiment or single-size experiment, this finding could be helpful in knowing whether the domain transport mode is saltation or rolling or sliding. However, for natural field conditions, it is very difficult to improve the particle-size identification by removing the effect of transport mode on the signal responses of the SPG system, due to fact that the signal amplitude shows dependency on both particle size and on transport mode. Some statements have been added to the main text (L433-436).

To improve the calibration curves between Amp$_{\text{Max,Pac}}$ and particle size $D$ by removing the influence of transport mode, some other information should be combined, which needs to be further studied.

● **Comment 19:** L400-441: Finally, can we conclude that the centroid frequency is a good proxy for size identification? [few dependency on transport mode and on particle velocity I guess].

**Response:** Controlled flume experiments showed that the centroid frequency is less sensitive to varying flow velocities than the maximum packet amplitude (Wyss et al. 2016, WRR). In this study, we found that the dependences of centroid frequency on transport mode and impact angle are less than those of maximum amplitude on transport mode and impact angle. Therefore, we could say that the centroid frequency is somewhat better suited for particle size identification than the maximum amplitude (L476-480).

Furthermore, the information of frequency and amplitude can be combined to further improve the accuracy of particle size identification, which has been developed by Nicollier et al. (2021b, in review), as we have introduced in section 2.4.2 of the present paper.

**Additional clarifications**

In addition to the above comments, all spelling and grammatical errors pointed out by the reviewer have been corrected in the manuscript.

We look forward to hearing from you in due time regarding our submission and to respond to any further questions and comments you may have.

Sincerely,
The authors of manuscript esurf-2021-72.

---

## Author Response (AR2)

Dear Editor Rebecca Hodge,

We appreciate your valuable comments on this manuscript. All the comments are helpful to further improve the manuscript. Some revisions have been made and highlighted (in red) in the revised manuscript based on these comments.

The following are point-by-point responses to the comments and suggestions, that we hope will meet with your approval. In our responses, the line number citations refer to the revised version of the manuscript.

**Comments from Editor**

- **Comment 1:** Thanks for undertaking a thorough review of your paper, and for clearly addressing all the reviewers' comments. I have some minor suggestions for places where I think that the paper would benefit from additional clarification (listed below). I recommend the paper for publication subject to these minor revisions.
  Comments by line numbers (in the tracked changes version of the text).

  **Response:** Thanks for your valuable comments on our manuscript. The responses to your comments are presented as follows.

- **Comment 2:** Abstract: The abstract does not mention the use of the FEM, which should be included.

  **Response:** Agreed. We added the following sentence in the abstract (L12-13):

  "The finite element method (FEM) was utilized to construct a numerical model of the SPG system and to simulate the signals triggered by a quartz sphere hitting the plate center with impact angles ranging from 0° to 90°."

- **Comment 3:** 66: Not all readers will be familiar with the finite element method, so explain briefly what it is.

  **Response:** Agreed. We added the following statements to explain the finite element method:

  "The dynamic response of the SPG system that corresponds to the recorded signal can be fully described by the partial differential equations (PDEs) based on elastoplastic mechanics, and these PDEs can be numerically solved by the FEM formulations resulting in a system of algebraic equations."

  See lines 69-71 in the revised manuscript.

- **Comment 4:** 300: Without referring back to your definitions, the differences between 'real', 'apparent' and 'effective' impacts aren't obvious to me. I can't think of better terms though. Maybe remind the reader of your definitions here. I'm not sure that 'effective' is the right term for the sum of real and apparent impacts. How about 'total recorded' impacts? Or given that the term is only used in a couple of places, just use 'real and apparent impacts?

  **Response:** Thanks for this valuable suggestion. The packets triggered by bedload particles impacting on the SPG plate above the considered geophone sensor are being classified as "real". The packets triggered by bedload impacts on a neighboring plate or on the concrete bed of the flume are classified as "apparent". We think the term "total recorded impacts" is an accurate way to describe the sum of real and apparent impacts This has been revised accordingly in the manuscript (L303-306, 308, 312).

● **Comment 5:** Figure 6: Explain in the caption how transport mode was determined for these data. It might also help to recap the difference between real/apparent/effective impacts.

**Response:** Some related statements have been added in the figure caption (L311-316).

● **Comment 6:** Table 5: It's not clear to me why you've included the 25th and 75th percentiles, but not the median or mean.

**Response:** Here we just want to show the range or variability of these parameters.

● **Comment 7:** 373: What is a non-effective impact?

**Response:** A non-effective impact here means a particle impact that does not trigger any signal packets. In other words, an impact that cannot be detected by our system. To make these sentences moreclear, we rephrased them as follows, avoiding the term non-effective impact (L377-380):

"The ratio $r_{i,j}^{Packet,V\_F}$ is close to one for two reasons. First, in the video analysis, we considered only the signal packets that were generated by particle impacts on the SPG plates. Second, given that impacts of such small particles are generally too weak to generate apparent packets, the number of detected packets can be expected to be close to the number of impacts on the SPG plates. A possible explanation for $r_{i,j}^{Packet,V\_F} < 1$ could therefore be the limited visibility during the video analysis due to flow turbulence, resulting in an underestimation of the number of impacts on the SPG plates."

● **Comment 8:** 489: I don't follow the sentence that starts: 'Auel et al...'. Please rephrase.

**Response:** In Auel et al. (2017a), the definition of the probability for the rolling mode is stated as follows:

"*The probability $P_R$ is defined as the distance covered by a particle in rolling motion divided by its total covered distance given by the sum of rolling and saltation motion averaged over n particle travels.*"

With regard to this definition, we know that $P_R$ is defined as a ratio of two travelled distances. One (numerator) is the travelled distance covered by a rolling particle. The other (denominator) is given by the travelled distance covered by both saltation and rolling modes (or say all transport modes because Auel et al. did not distinguish the rolling and sliding modes) that is averaged over the numbers of particles.

Hence, we rephrased the statement as follows (L494-495):

"Auel et al. (2017a) defined the probability for the rolling mode as the ratio of the travelled distance covered by a rolling particle to the overall travelled distances determined by the sum of all transport modes that are averaged over numbers of particles."

● **Comment 9:** Figure 12: What is the pink zone in panel b? Is this the red shaded area referred to in line 518? It needs to be included in the legend and/or caption.

**Response:** It is the red shaded area with 75% transparency, so it may look a little pink. This area indicates variabilities of the particle velocity $V_P^{Est,*}$ estimated from the flow velocity, ranging from 30% to 80% as suggested by Julien and Bounvilay (2013).

We have added a sentence in the figure caption of Fig. 12b in the revised manuscript (L536-537).

**Additional clarifications**

In addition to the above comments, all spelling and grammatical errors have been checked and corrected in the manuscript.

We look forward to hearing from you in due time regarding our submission and to respond to any further questions and comments you may have.

Sincerely,

The authors of manuscript esurf-2021-72.